# Cell-guiding microporous hydrogels by photopolymerization-induced phase separation

Monica Z. Müller[1,5], Margherita Bernero [1,5], Chang Xie[1,2], Wanwan Qiu[1,2], Esteban Oggianu [1], Lucie Rabut [1], Thomas C. T. Michaels [2,3], Robert W. Style [4], Ralph Müller [1] & Xiao-Hua Qin [1,2] ✉

Microporous scaffolds facilitate solute transport and cell-material interactions, but materials allowing for spatiotemporally controlled pore formation in aqueous solutions are lacking. Here, we propose cell-guiding microporous hydrogels by photopolymerization-induced phase separation (PIPS) as instructive scaffolding materials for 3D cell culture. We formulate a series of PIPS resins consisting of two ionic polymers (norbornene-functionalized polyvinyl alcohol, dextran sulfate), di-thiol linker and water-soluble photoinitiator. Before PIPS, the polymers are miscible. Upon photocrosslinking, they demix due to the increasing molecular weight and form a microporous hydrogel. The pore size is tunable in the range of 2-40 μm as a function of light intensity, polymer composition and molecular charge. Unlike conventional methods to fabricate porous hydrogels, our PIPS approach allows for in situ light-controlled pore formation in the presence of living cells. We demonstrate that RGD-functionalized microporous hydrogels support high cell viability (>95%), fast cell spreading and 3D morphogenesis. As a proof-of-concept, these hydrogels also enhance the osteogenic differentiation of human mesenchymal stromal cells, matrix mineralization and collagen secretion. Collectively, this study presents a class of cell-guiding microporous hydrogels by PIPS which may find applications in complex tissue engineering.

Microporous scaffolds[1-3] have emerged as promising materials for 3D cell culture and tissue engineering since they facilitate solute transport, cell-cell communication, and tissue ingrowth. Methods to create micropores in the range of 1–200 μm include porogen leaching[1-3], microgel annealing[4-7], microstrand molding[8], phase separation through emulsification[9,10] or polymerization[11-13], and laser erosion[14,15]. For instance, Huebsch et al. created microporous alginate hydrogels by adding hydrolytically degradable porogens for recruiting or releasing cells[2]. Since hydrolytic degradation is a slow process, the embedded cells need at least 7 days to colonize and interact with the void space. Ying et al. [9] reported aqueous two-phase systems to engineer microporous polyethylene glycol (PEG)-gelatin methacryloyl hydrogels. Although photocrosslinking is applied to arrest two distinct phases in an emulsion, this process does not allow for spatiotemporal control of pore formation.

A major challenge in additive (bio)manufacturing[16-19] is the shortage of bioresin formulations that allow facile fabrication of hydrogel constructs with interconnected cell-scale microporosities as guidance cues for 3D cell growth and tissue regeneration. Existing hydrogel systems[17,20] for 3D bioprinting often have nanopores with

[1]Institute for Biomechanics, ETH Zurich, Zurich, Switzerland. [2]Bringing Materials to Life Initiative, ETH Zurich, Zurich, Switzerland. [3]Institute of Biochemistry, ETH Zurich, Zurich, Switzerland. [4]Laboratory for Soft Materials and Interfaces, ETH Zurich, Zurich, Switzerland. [5]These authors contributed equally: Monica Z. Müller, Margherita Bernero. ✉e-mail: qinx@ethz.ch

very small sizes (5–100 nm) that fail to provide a permissive environment for encapsulated cells. A combination of 3D printing and phase separation has been explored in other materials than hydrogels, such as glass[21] and acrylic monomers[22,23] using digital light processing[21,22] and two-photon polymerization[23]. However, these methods are incompatible with biofabrication processes in the presence of living cells.

Liquid–liquid phase separation has raised increasing attention in life sciences owing to its vital role in human health and diseases[24,25]. Phase separation in living systems relies on macromolecules. Unlike small molecules, macromolecules tend to separate and form droplets in aqueous solutions when the mixing entropy is not favorable due to their large molar mass[12,26]. Herein, we report fast construction of cell-guiding microporous hydrogels by leveraging a thiol-ene photoclick resin[18,27] and photopolymerization-induced phase separation (PIPS, Fig. 1a). In PIPS, photopolymerization induces phase separation, driven by changes in entropy as the molecular weight of polymers increases during in situ photocrosslinking. This process can be used to form microporous structures within a hydrogel (Fig. 1b, c). We employ PIPS to produce cell-compatible microporous hydrogels within seconds via efficient thiol-ene photoclick polymerization (Fig. 1d, e). Before photocuring, the resin is optically transparent. After PIPS, the materials form stable microporous hydrogels in the presence of living cells. The objective of this study is to screen the formulations suitable for PIPS and investigate how fine-tuned microporous structures in PIPS hydrogels influence cell spreading and 3D morphogenesis over time compared to conventional nanoporous hydrogels.

## Results and discussion
### Design considerations of a PIPS resin
In the present study, we devised an efficient photoclick PIPS resin based on the following considerations. First, we focused on the properties of aqueous mixtures of two nonionic polymers: polyvinyl alcohol (PVA) and dextran. Aqueous solutions of these two polymers are known to demix as a function of temperature and salt additives

when the mixing entropy is unfavorable[28]. We chose to develop a photoclick resin composed of two ionic synthetic polymers that are miscible in water before photocrosslinking but undergo in situ phase separation upon exposure to light. We previously reported a synthetic polymer based on norbornene-functionalized PVA (nPVA)[27], which is water soluble and highly efficient for hydrogel formation in the presence of thiols via step-growth thiol-ene photoclick polymerization. This process can be spatiotemporally controlled by light using a water-soluble photoinitiator such as LAP (Fig. 1d)[29]. Thus, we reasoned that aqueous mixtures of nPVA and an ionic derivative of dextran may undergo PIPS to form microporous hydrogels in a spatially and temporally controlled fashion. We selected dextran sulfate (DS) as a non-crosslinkable polymer due to its ionic nature and commercial availability. Second, the resin formulations must be cell-friendly to enable in situ photoencapsulation of living cells within hydrogels by means of state-of-the-art 3D cell culture techniques.

### Evidence of PIPS
To determine potential compositions for PIPS, we constructed an approximate phase diagram of nPVA ($M_W$: 54 kDa, degree of functionalization: 3.5%) and DS ($M_W$: 40 kDa), as shown in Fig. 2a. Despite the modifications to the polymers, this diagram resembles a phase diagram of PVA and dextran reported in literature[28]. We selected compositions near the critical point of the binodal, specifically between 2–3.5% nPVA and 2–3.5% DS. This strategy ensured that the phases maintained nearly equal volumes and remained close to the binodal, which is crucial for achieving the controlled PIPS necessary to obtain the desired pore morphology. Notably, the binodal shifts to smaller concentrations with an increase in molecular weight (Supplementary Fig. 1). We performed in situ photo-rheometry to evaluate the crosslinking of the compositions. The results showed rapid crosslinking within 30 s, as indicated by the sharp increase of the storage modulus (G′) upon exposure to light (Fig. 2b). After PIPS, the hydrogels became turbid presumably due to the increase of light scattering (Fig. 2c).

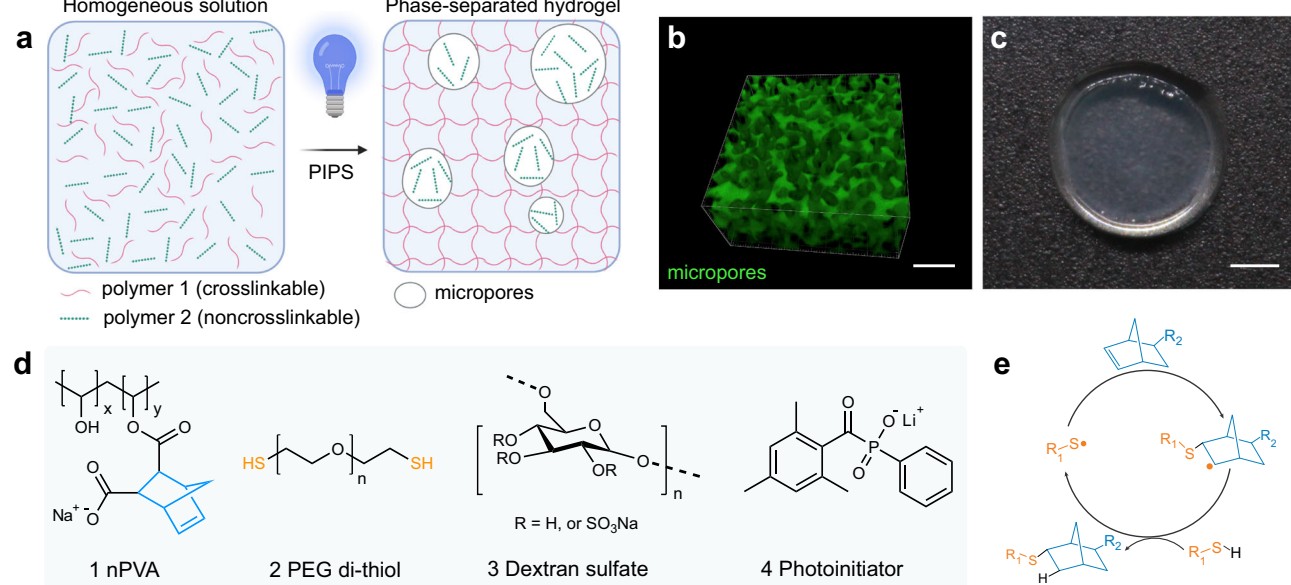

**Fig. 1 | Microporous hydrogels by photopolymerization-induced phase separation (PIPS). a** Schematic representation of the PIPS process: During photocrosslinking, the increasing molecular weight of the crosslinkable polymers drives an entropy change that leads to the separation of one phase from two water-soluble polymers into two distinct phases. Pore formation is a result of phase separation and nucleation into droplets which get arrested by the stiffening polymer network. Created in BioRender. Bernero, M. (2025) https://BioRender.com/

9r5esrt. **b** Representative confocal microscopy image of a microporous hydrogel permeabilized with FITC-dextran. Scale bar = 10 μm. **c** Photograph of a phase-separated hydrogel. Scale bar = 2 mm. **d** Chemical structures of key components for PIPS: (1) norbornene-functionalized polyvinyl alcohol (nPVA), (2) PEG-di-thiol (PEG-2-SH) linker, (3) dextran sulfate and (4) water-soluble photoinitiator (LAP, lithium phenyl-2,4,6-trimethylbenzoylphosphinate). **e** Schematic of the proposed mechanism in radical-mediated thiol-norbornene photoclick polymerization.

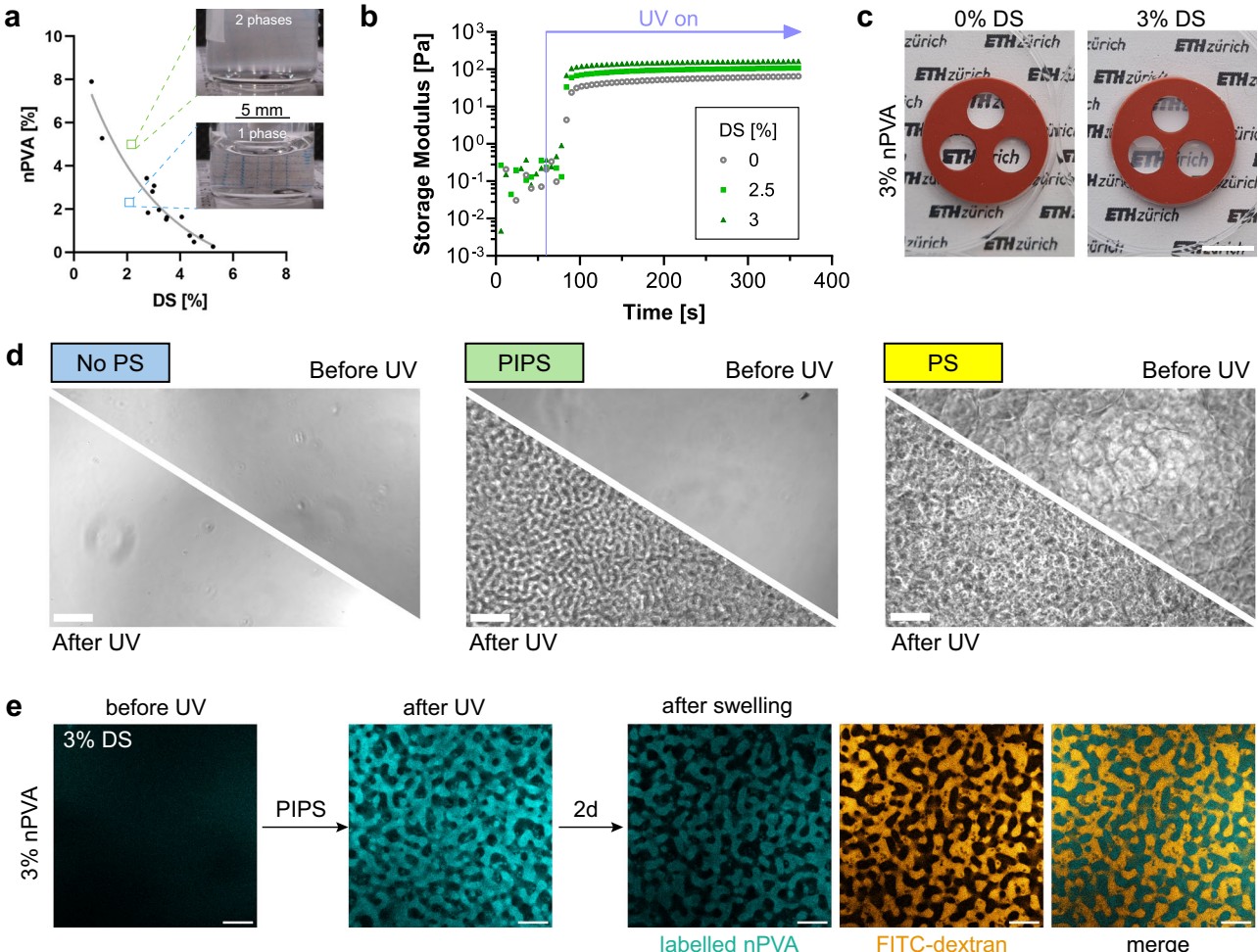

**Fig. 2 | Microporous hydrogels by PIPS between norbornene-functionalized PVA (nPVA) and dextran sulfate (DS). a** Approximated phase diagram between nPVA ($M_W$: 54 kDa) and DS ($M_W$: 40 kDa). **b** Time sweep of storage (G′) and loss (G″) modulus of 2% nPVA with 0, 2.5 and 3% DS under UV irradiation after 60 s. **c** Pictures of hydrogels with 3% nPVA and 0% DS (no PIPS) or 3% DS (PIPS, turbid) in a PDMS mold after UV. **d** During screening for polymer mixtures undergoing PIPS upon UV irradiation (Table 1), the following behaviors were observed: No phase separation (No PS)—a homogeneous mixture before and after polymerization; Photopolymerization-induced phase separation (PIPS)—phase separation occurs during polymerization; and Phase separation (PS)—phase separation occurs prior to polymerization, resulting in distinct phases. Representative images of each behavior are shown, illustrating the corresponding phase transitions in the mixture. **e** Representative confocal images of 3% nPVA + 3% DS before and after UV crosslinking, and after swelling. The hydrogel was labeled fluorescently with rhodamine-MA (cyan) and 500 kDa FITC-dextran was diffused into the pores (orange) after swelling. Images from the middle of the gels demonstrate that the FITC-dextran dye penetrates through the interconnected micropores and occupies the pore space, complementing the labeled hydrogel phase. Scale bars: 5 mm (**c**), 100 µm (**d**), 20 µm (**e**).

Next, we experimentally screened the feasibility of different resin compositions for PIPS on an optical microscope (Fig. 2d and Table 1). Phase-contrast images were taken before and after UV curing. When both nPVA and DS are below 2.5%, no phase separation was observed before or after UV. At higher concentrations, PIPS occurred as evidenced by the transition from an optically clear mixture (before UV) to two phases (after UV). However, further increases of either nPVA or DS resulted in emulsions where the compositions demixed before UV.

To investigate the phase behavior of compositions undergoing PIPS, rhodamine B methacrylate (rhodamine-MA) was added to the resins as a fluorescent reporter of phase separation, as it is conjugated to the nPVA-PEG phase during photocrosslinking. The samples before and after UV curing were imaged by confocal microscopy. As shown in Fig. 2e, porous structures were observed only after UV curing. This observation confirms that the gel precursors were initially miscible, and phase separated due to photocrosslinking. The radial intensity plot of the Fast Fourier Transformation (RIFFT) of the image after curing shows a clear peak, implying a uniform length scale

(Supplementary Fig. 2a). No phase separation was found in the samples with DS concentrations below 2.5% and without DS (Supplementary Fig. 3). Since the pore size may change after swelling, imaging was repeated after 2 days. The pore length scale for the 3% DS group did not change significantly, although small changes were evidenced in another composition (Supplementary Fig. 2b, c). A dye diffusion experiment was performed to verify the gel microstructure. Confocal microscope imaging data showed that FITC-dextran ($M_W$: 500 kDa) could permeate the pore space (Fig. 2e and Supplementary Movie 1). No pores were seen in the sample without DS (Supplementary Fig. 4). These results suggest that the pores in PIPS hydrogels are permeable by FITC-dextran tracers.

### The impact of resin composition on PIPS: pore size, gel mechanics, and turbidity

To study the effect of molecular charge on PIPS, the PIPS resin was compared to a composition with the nonionic dextran instead of DS. With the same concentration, no pores were seen in the sample with

**Table 1 | Screening for polymer mixtures undergoing PIPS upon UV irradiation**

| nPVA [%] | | | | | | | | |
|---|---|---|---|---|---|---|---|---|
| 5 | | PIPS | PS | ○ | ○ | ○ | ○ | ○ |
| 4 | | PIPS | PIPS | PS | PS | PS | ○ | ○ |
| 3.5 | | No PS | PIPS | PIPS | PS | PS | PS | ○ |
| 3 | | No PS | No PS | PIPS | PIPS | PS | PS | ○ |
| 2.5 | | ○ | No PS | No PS | PIPS | PS | PS | ○ |
| 2 | | ○ | ○ | No PS | PIPS | PIPS | PS | PS |
| 1 | | ○ | ○ | ○ | No PS | No PS | No PS | PIPS |
| | | 1 | 2 | 2.5 | 3 | 3.5 | 4 | 5 |
| | **DS [%]** | | | | | | | |

○Mixtures far beyond the PIPS region were excluded from further screening.

Evaluation based on the observable behaviors shown in Fig. 2d: no phase separation (No PS), photopolymerization-induced phase separation (PIPS), and phase separation before polymerization (PS)

dextran, while pores were seen with DS (Fig. 3a). RIFFT analysis revealed that there is no peak in the sample with dextran, whereas a peak corresponding to a pore size of 5 µm was found in the sample with DS. These results suggest that a charged polymer (DS) is more effective in inducing phase separation during crosslinking. This enhanced performance may arise from its more efficient repulsion of negatively charged nPVA, unlike dextran with the same molecular weight. The ability of charged polymers to undergo water-assisted microphase separation has been recently reported[30].

Resins with a higher nPVA content were also explored. A resin containing 3% nPVA and 3% DS exhibited the largest pores after PIPS (Fig. 3b), with the peak of the RIFFT corresponding to a length scale of 15 µm. The pore length scale decreased to 7 and 3 µm for the 2.75% DS and 2.5% DS group, respectively. At 3.5% DS, the composition phase separated even before gel formation (Supplementary Fig. 5). Compositions with even higher nPVA concentrations were also screened, but the pores were either not interconnected or less homogeneously distributed (Supplementary Fig. 6), presumably due to phase separation before photopolymerization.

Photo-rheology was used to determine the physical properties of PIPS hydrogels as a function of different compositions. The G′ was higher with an increase of nPVA% with the same thiol-ene ratio (Fig. 3c, d). The mean values were 146, 1344, and 3186 Pa for 2, 3, and 5% nPVA, respectively. These data indicate an increase in crosslinking density as the polymer content increases.

Next, we tested if the variation of DS content leads to changes in gel stiffness (defined as the G′-plateau). Interestingly, the stiffness was significantly higher for the sample with 2.5% DS compared to the control without DS (Fig. 3e). One possible explanation is that, during PIPS, the formation of nPVA- and DS-rich partitions promotes localized aggregation, thereby increasing the crosslinking density within the nascent hydrogel phase. In this way, the stiffness reductions typically anticipated from pore formation may be counterbalanced in the fully formed hydrogel. However, the specific mechanism underlying this effect is not well understood and has been scarcely documented in the literature. Since phase separation and pore formation coincide with the liquid-to-solid transition of nPVA, further investigations are needed to elucidate how these concurrent processes ultimately shape the mechanical properties. The amplitude and frequency sweeps showed that a strain of 0.5% was within the linear viscoelastic region and the gels exhibited a solid behavior over a wide range of frequencies (Supplementary Fig. 7). Interestingly, the mass swelling ratio of 2% nPVA hydrogels with varying DS content showed no significant difference (Supplementary Fig. 8). Future work is needed to investigate the dynamics of solute transport and swelling in these microporous hydrogels.

We studied the influence of different compositions on gel turbidity on a plate reader. The turbidity was higher in hydrogels with

compositions close to the critical point. For instance, hydrogels with 2% nPVA and 2.5% DS showed a significantly higher level of turbidity compared to 5% nPVA gels with the same DS concentration (Fig. 3f). Additionally, the turbidity of formulations with 4% nPVA and varying DS before and after UV curing was measured. Before UV curing, there was no significant difference in the turbidity. After UV curing, the turbidity of hydrogels with 1.5% DS was significantly higher compared to 0.5% DS and 1% DS groups as well as all groups before UV curing (Fig. 3g). This indicates that the composition with 4% nPVA and 1.5% DS was initially miscible, but phase separated upon photopolymerization.

We further analyzed the microporous structure in PIPS hydrogels. Using a MATLAB script[31], we segmented the confocal z-stacks of labeled hydrogels (3% nPVA and 2.5–3% DS) and quantified 3D pore characteristics: porosity, pore size, and pore connectivity. The results showed that the porosity increases significantly with higher DS concentration (Fig. 3h). The obtained pore size distributions also correlate with the characteristic pore length scales (Fig. 3i, j) determined using the 2D RIFFT method. The 2.5% DS composition with the smallest length scale has the highest frequency of small pores in the range of 2–3 µm, while the compositions with higher DS concentration and larger pore length scales contain more pores with larger diameters up to 20 µm. Finally, the pore connectivity quantification verified that the pore space is interconnected (Fig. 3k). Most pores are directly connected to at least two other pores, ensuring continuity of the pore space. This property may promote solute transport and 3D cell growth throughout the hydrogels.

Bicontinuous structures in polymer networks have been generated via the competition between photopolymerization and phase separation[32]. Interestingly, we observed that for 3% nPVA hydrogels, the G′ first increased with the addition of DS, but dramatically decreased when reaching 3% DS (Table 2). Compared to the 2% nPVA compositions, the 3% nPVA composition is closer to the binodal curve at the onset of polymerization. As the phase separation proceeds, the kinetics of PIPS may differ from compositions of lower polymer content, and the effective light intensity may be altered. Future work is needed to investigate how crosslinking and phase separation are correlated using photo-rheology coupled with time-lapsed pore imaging.

### The impact of light intensity on PIPS

Tuning the pore size by light intensity will significantly expand the applicability of PIPS hydrogels for bio-applications. Such tuning has been only shown in purely organic polymeric matrices in the literature[32]. Thus, we investigated the effect of light intensity (5–100 mW cm$^{-2}$) on PIPS and pore size distribution. As anticipated, confocal imaging of PIPS hydrogels showed that the length scale of the pores can be tuned by UV light intensity (Fig. 4a). For irradiation at an intensity of 5 mW cm$^{-2}$, the pore size was 8 µm. As the light intensity increased to 10 mW cm$^{-2}$, the pore size decreased to 5 µm. Further, it

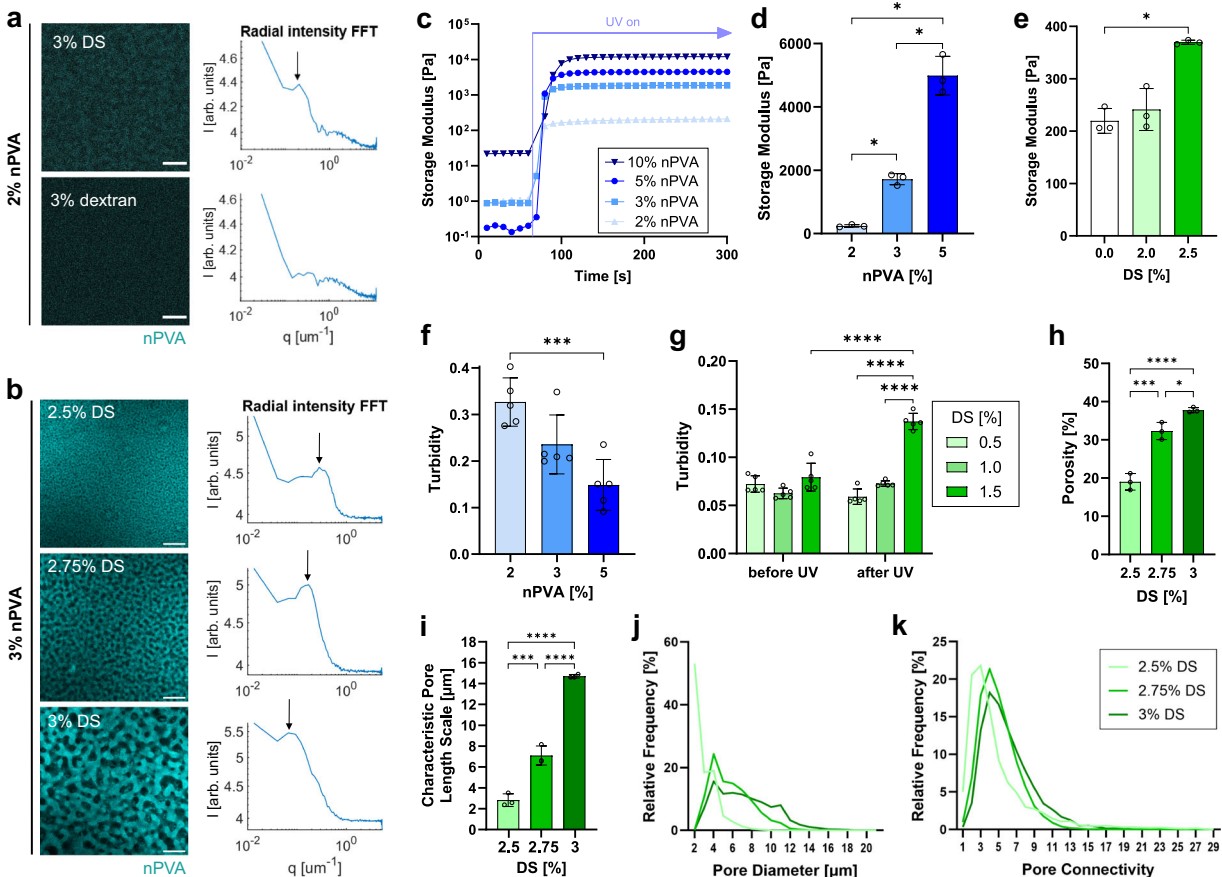

**Fig. 3 | The effect of material composition on structural and physical properties of PVA hydrogels. a** Molecular charge promotes the phase separation between nPVA and DS. Representative confocal images of the compositions containing 2% nPVA and 3% DS/dextran after UV curing. FITC-labeled nPVA (cyan) was used for confocal imaging. RIFFT of the images showing a clear peak with DS. **b** Phase separation with 3% nPVA and different DS concentrations, resulting in pores at different scales. Representative confocal images of hydrogels labeled with rhodamine-MA after UV curing and RIFFT showing clear peaks (arrows) at different length scales. **c**–**e** In situ photo-rheometry of hydrogels with varying nPVA (**c**, **d**) and DS (**e**) concentrations; 10 rad s⁻¹ and 0.5% strain. **c** Representative time sweep of $G'$ of gels with 2% DS and varying nPVA content under UV after 60 s. **d**, **e** $G'$ of gels with either 2% DS (**d**) or 2% nPVA (**e**) after 4 min of crosslinking ($N = 3$ samples, Brown-Forsynthe and Welch ANOVA/Dunnett's T3); $p$ values: $p = 0.0102$ (2 vs. 3% nPVA), $p = 0.0115$ (2 vs. 5% nPVA), $p = 0.0258$ (3 vs. 5% nPVA), $p = 0.0177$ (0 vs. 2.5%

DS). **f**, **g** Gel turbidity is higher in phase-separated compositions close to the critical point ($N = 5$ samples, one- and two-way ANOVA/Tukey for **f** and **g**, respectively). **f** Turbidity of cured 2% nPVA gels with 2.5% DS showing significantly higher turbidity than 5% nPVA; ***$p = 0.0009$. **g** Turbidity of 4% nPVA gels with 0.5, 1, and 1.5% DS before and after UV curing; ****$p < 0.0001$. **h**–**k** Pore quantification of compositions with 3% nPVA and different DS concentrations. **h** Porosity of hydrogels determined through 3D pore analysis ($N = 3$ samples, one-way ANOVA/Tukey). *$p = 0.0255$, ***$p = 0.0003$, ****$p < 0.0001$. **i** Characteristic pore length scale obtained from RIFFT plot peaks of such images ($N = 3$ samples, one-way ANOVA/ Tukey). ***$p = 0.0004$, ****$p < 0.0001$. **j** Pore size distribution was determined from the external radii in 3D pore analysis. Pore sizes below 1.5 μm were excluded due to the resolution limit. Bin size = 1. **k** Pore connectivity was defined as the number of directly adjacent pores and was quantified for all the pores included in (**i**). Bin size = 1. Scale bars: 10 μm (**a**) and 20 μm (**b**). Data presented as mean ± SD.

decreased to 2 μm when the light intensity increased to 100 mW cm⁻² (Fig. 4b). To understand this effect, we devised a simple model describing how pore size $\ell$ scales with light intensity $I$. In this model, we assume that polymer domains form initially via PIPS (either via spinodal decomposition or nucleation and growth) and coarsen over time. Eventually, domain growth is arrested at the onset of gelation, which effectively freezes the structure and sets the final pore size. The characteristic size of pores is thus set by a balance of PIPS dynamics and crosslinking kinetics.

Regarding crosslinking, photo-rheology measurements showed that an increase in light intensity accelerates crosslinking (Fig. 4c–f). The evolution of $G'$ with different UV light intensities is shown in Fig. 4c. For analyzing the rate of crosslinking, linear regression was performed on the $G'$ plots. The slope, which indicates the rate of crosslinking, was higher with increased light intensity (Fig. 4d). The delay time denotes the timepoint where the linear regression curve starts (Fig. 4e), whereas T₈₀% refers to the time to reach 80% of the

maximum $G'$ within 5 min of UV curing. When increasing the light intensity from 5 to 20 mW cm⁻², the crosslinking speed is increased from 1.8 to 8.8 Pa s⁻¹, whereas the delay time is decreased from 100 to 30 s, and the time to reach T₈₀% is decreased from 198 to 68 s, respectively. Overall, our data indicate a simple first-order kinetics model for the rate of crosslinking with light intensity $R_c \sim I$. Thus, the characteristic timescale of crosslinking scales with light intensity as $\tau_c \sim R_c^{-1} \sim I^{-1}$. The results confirmed that photocrosslinking is faster at higher light intensities (Fig. 4f). With higher light intensity, the pore morphology has less time to evolve before it is arrested at the onset of gelation. Assuming diffusive coarsening, the characteristic size of the PIPS domains grows with time as $\ell \sim t^{1/3}$. Thus, within the characteristic time of crosslinking $\tau_c$, the phase-separated domains will grow to a size $\ell \sim I^{-1/3}$, which is the characteristic length scale of the pores. Note that at lower light intensity, when the PIPS domains have more time to grow before gelation, domain growth can transition from diffusive to viscous coarsening. In viscous coarsening, the

**Table 2 | Composition and properties of common mixes used in this study**

| Mix No. | nPVA % | DS % | Light int. [mW/cm²] | Pore size [µm]ᵃ | G' [Pa]ᵉ | G" [Pa]ᵉ | G"/G'ᵉ |
|---|---|---|---|---|---|---|---|
| 0 | 2.0 | - | 20 | NAᶜ | 62.5 ± 2.0 | 1.1 ± 0.07 | 0.018 ± 0.001 |
| 1 | 2.0 | 2.5 | 20 | 3 | 111.4 ± 5.0 | 0.68 ± 0.12 | 0.006 ± 0.001 |
| 2 | 2.0 | 3.0 | 20 | 7 | 162.2 ± 3.6 | 0.75 ± 0.03 | 0.005 ± 0.0002 |
| 3ᵇ | 2.0 | 1.0 | 20 | 18 | 4249 ± 1063ᵈ | 94.2 ± 38.9ᵈ | 0.022 ± 0.004ᵈ |
| 4 | 3.0 | - | 10 | NAᶜ | 1474 ± 281.2 | 2.95 ± 0.57 | 0.002 ± 0.0003 |
| 5 | 3.0 | 2.5 | 10 | 3 | 1755 ± 472.5 | 6.81 ± 2.59 | 0.004 ± 0.001 |
| 6 | 3.0 | 3.0 | 10 | 14 | 242.3 ± 51.30 | 4.50 ± 1.55 | 0.019 ± 0.008 |

ᵃDetermined by RIFFT of confocal imaging data after gel swelling.
ᵇ5% gelatin was added to enhance the viscosity of the resin.
ᶜNo pores detected.
ᵈBefore incubation at 37 °C (prior to gelatin release).
ᵉG', G", and G"/G' are the plateau values after 5 min of UV irradiation.
0.05% LAP, UV-365nm light irradiation, PEG-2-SH crosslinker for a thiol:ene ratio of 1.6

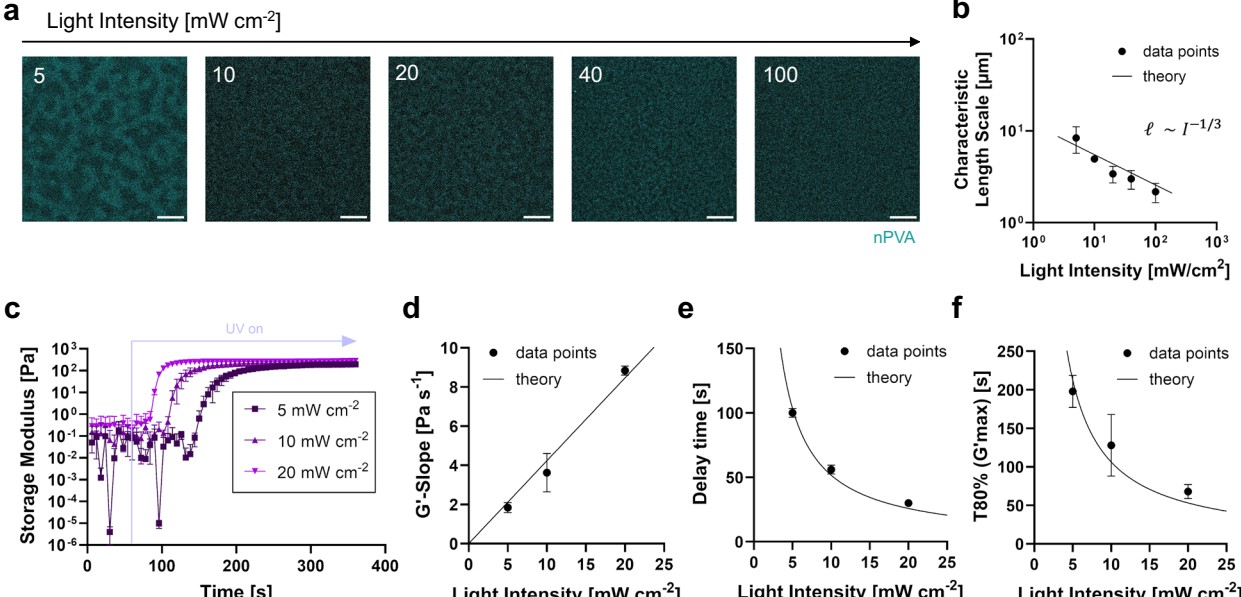

**Fig. 4 | The effect of light intensity on pore size and photocrosslinking dynamics. a, b** Characteristic length scale of the pores can be tuned as a function of the light intensity. **a** Representative confocal images of UV-cured hydrogels (2% nPVA-3% DS) with light intensities ranging from 5 to 100 mW cm⁻² (365 nm). FITC-labeled nPVA was included for imaging. **b** Characteristic length scale obtained from RIFFT plot peaks of such images (*N* = 3 samples); curve fit according to the theoretical model. **c–f** Crosslinking dynamics with different light intensities: effect on the time sweep of gel storage moduli (**c**) and G'-slope (**d**), delay time for crosslinking (**e**) and time to reach 80% of the G'-plateau (**f**) under UV irradiation from 60 s at different light intensities (5–20 mW cm⁻²); curve fits according to the theoretical model (*N* = 3 samples). Data presented as mean ± SD. Scale bars: 10 µm (**a**).

characteristic size of PIPS domains scales with time as $\ell \sim t$, which gives $\ell \sim I^{-1}$ [33,34]. Thus, at lower light intensity, we expect a transition for the pore size scaling from $\ell \sim I^{-1/3}$ to $\ell \sim I^{-1}$. Kimura et al. investigated the process of PIPS in a mixture of polystyrene and methylacrylate [35]. By changing the light intensity, they obtained a variety of stationary morphologies. Together, these findings suggest that the light intensity can be used to tune the crosslinking dynamics and PIPS, which eventually influence the length scale of the pores.

### 3D osteogenic culture of hMSCs

For studying 3D cell-material interactions, human mesenchymal stromal cells (hMSCs) were photoencapsulated in RGD-functionalized hydrogels (PIPS: 3% nPVA and 3% DS; control: 3% nPVA) at a density of $4 \times 10^6$ cells/mL. After 5 min of UV irradiation, cells in both compositions showed a high viability (>90%) on day 1. But viability remained

significantly higher in the PIPS group over the course of a week (Fig. 5a, b). On day 7, viability was >98% in the PIPS group, whereas it dropped significantly in the control.

Actin-nuclei-staining results show that cells in the PIPS hydrogels exhibited a higher extent of spreading and intercellular connectivity compared to the control (Fig. 5c). Fast dendritic outgrowth into the micropores within 24 h was evidenced by co-staining of the pore space using a FITC-dextran tracer (Fig. 5d and Supplementary Movie 3). Cells in the PIPS hydrogels occupy a significantly larger area compared to the control on day 1, 7, and 28 (Fig. 5e). Still, the confocal images showed pronounced dendritic cell processes in the PIPS gels at all timepoints over the course of 28 days (Supplementary Fig. 9 and Supplementary Movie 2). In contrast, cells in the control hydrogels exhibited elongated rather than stellate morphologies. This difference is attributed to the increase in pore space, which was confirmed by FITC-dextran perfusion (Supplementary Fig. 9b).

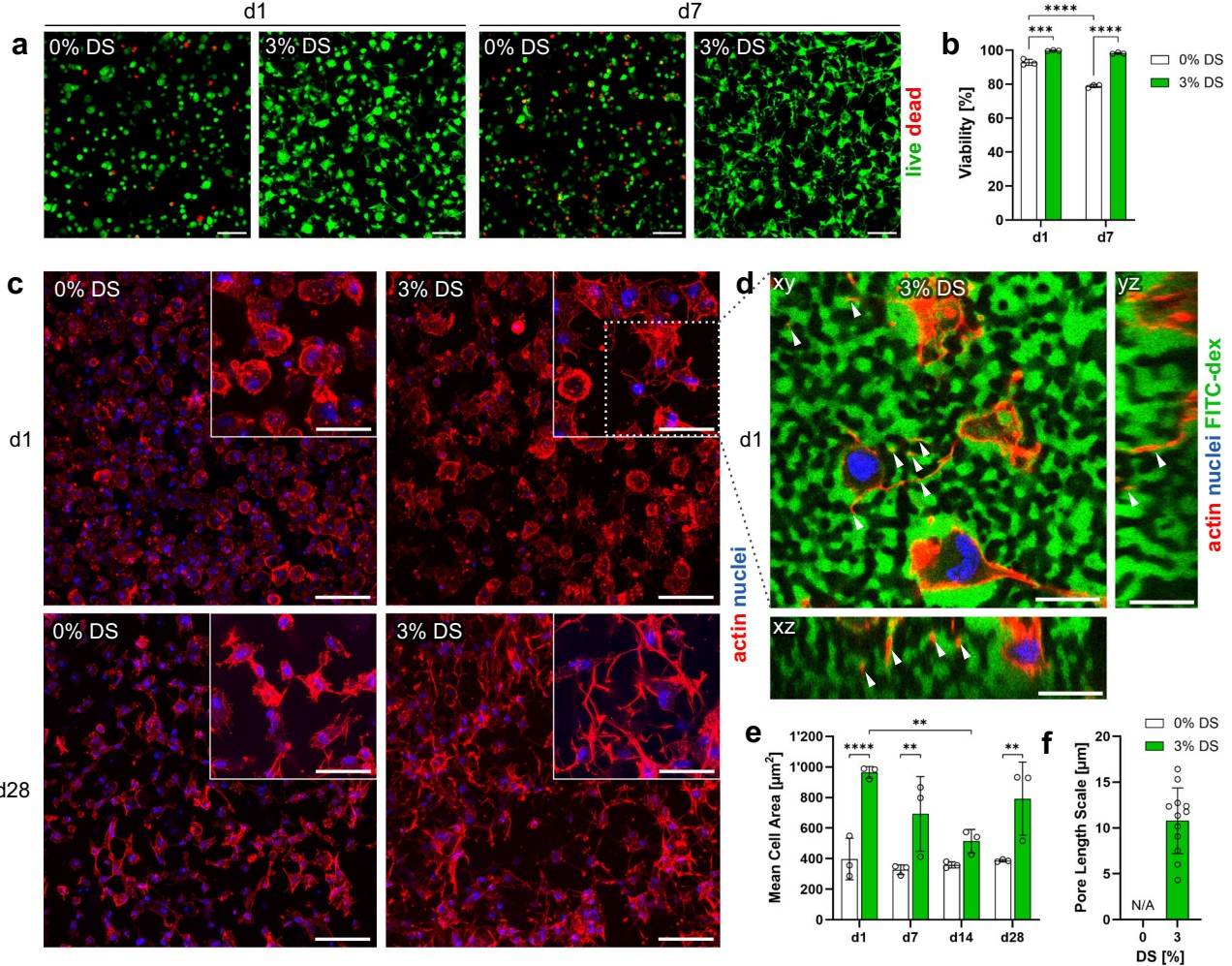

**Fig. 5 | 3D photoencapsulation of hMSCs in microporous PVA hydrogels.**
**a** Representative maximum intensity projections of hMSCs encapsulated in the PVA hydrogels, stained with calcein AM (live) and ethidium homodimer 1 (dead) on days 1 and 7 in osteogenic culture. **b** Cell viability on day 1 and 7 as quantified from the live-dead stained samples ($N = 3$ samples, two-way ANOVA/Tukey). ***$p = 0.0006$, ****$p < 0.0001$. **c** Representative maximum intensity projections of embedded hMSCs stained with Hoechst (nuclei) and phalloidin (actin) on days 1 and 28. All other timepoints can be found in Supplementary Fig. 9. **d** An orthogonally sectioned confocal z-stack on day 1, showing how cells' dendritic processes (arrows)

penetrate the pores (green, perfused with FITC-dextran) within a 3% DS hydrogel. **e** Mean cell area after 1, 7, 14, and 28 days of 3D osteogenic culture ($N = 3$ samples, two-way ANOVA/Šidák). $p = 0.005$ (3% DS d1 vs. d14), $p < 0.0001$ (d1 0 vs. 3% DS), $p = 0.0043$ (d7 0 vs. 3% DS), $p = 0.002$ (d28 0 vs. 3% DS). **f** Characteristic length scale of the pores in the cellular hydrogels obtained from RIFFT plot peaks of FITC-dextran-perfused pore images ($N = 12$ samples at different timepoints, see also Supplementary Fig. 10). Data presented as mean ± SD. Scale bars: 100 μm (**a**, **c**), 50 μm (**c**, inserts), and 20 μm (**d**).

We further tested whether cell inclusion may interfere with pore structure in PIPS. We performed 2D RIFFT analysis with the cellular samples and confirmed the formation of pores with a uniform length scale in the PIPS hydrogels (Supplementary Fig. 10a–c). However, we observed a larger variation in the pore length scale than acellular samples (Fig. 5f). When analyzing the 3D pore characteristics, we find that porosity and pore connectivity remain consistent throughout the samples and timepoints, while the variation in pore sizes is confirmed (Supplementary Fig. 10d–f).

We then investigated whether the PIPS hydrogels can support 3D osteogenic differentiation of hMSCs in a proof-of-concept study. Following a 28-day culture, a greater extent of osteocalcin (Ocn), a marker for osteoblasts, was found in the PIPS hydrogels (Fig. 6a, b). The expression of alkaline phosphatase (ALP) was quantified from the 3D culture samples using a colorimetric assay (Fig. 6c). ALP activity increases over time in the PIPS hydrogel, while it decreases to below baseline (day 0) for the control, resulting in significant differences after one week of osteogenic culture. Additionally, we evaluated mineral (hydroxyapatite) deposition using an OsteoImage staining

assay (Fig. 6d, e). While signals on day 1 were negligible, mineralization was observed after 28 days. The mineralized area in the PIPS hydrogel was more than twofold larger than in the control. Although differences in solute transport may influence mineralization, we reason that the greater extent of mineralization in the PIPS group is due to the cells themselves, as acellular controls show no signal (Supplementary Fig. 11b).

### 3D culture of human fibroblasts
Besides 3D hMSC cultures, we cultivated human dermal fibroblasts (HDFs) in the PIPS and control hydrogels (Fig. 7a and Supplementary Fig. 12). Early changes in cell morphology were determined within 48 h. Significant cell spreading was observed as early as 12 h in the PIPS group. Cells continued 3D morphogenesis over 48 h (Fig. 7a). The control group showed minimal cell spreading, with only a slight increase at 48 h. Quantitative analysis of the total cell spreading area confirmed these findings (Fig. 7b). Interestingly, when grown in the PIPS hydrogels without RGD, cells remained round with almost no spreading (Fig. 7c, d). These findings demonstrate that PIPS

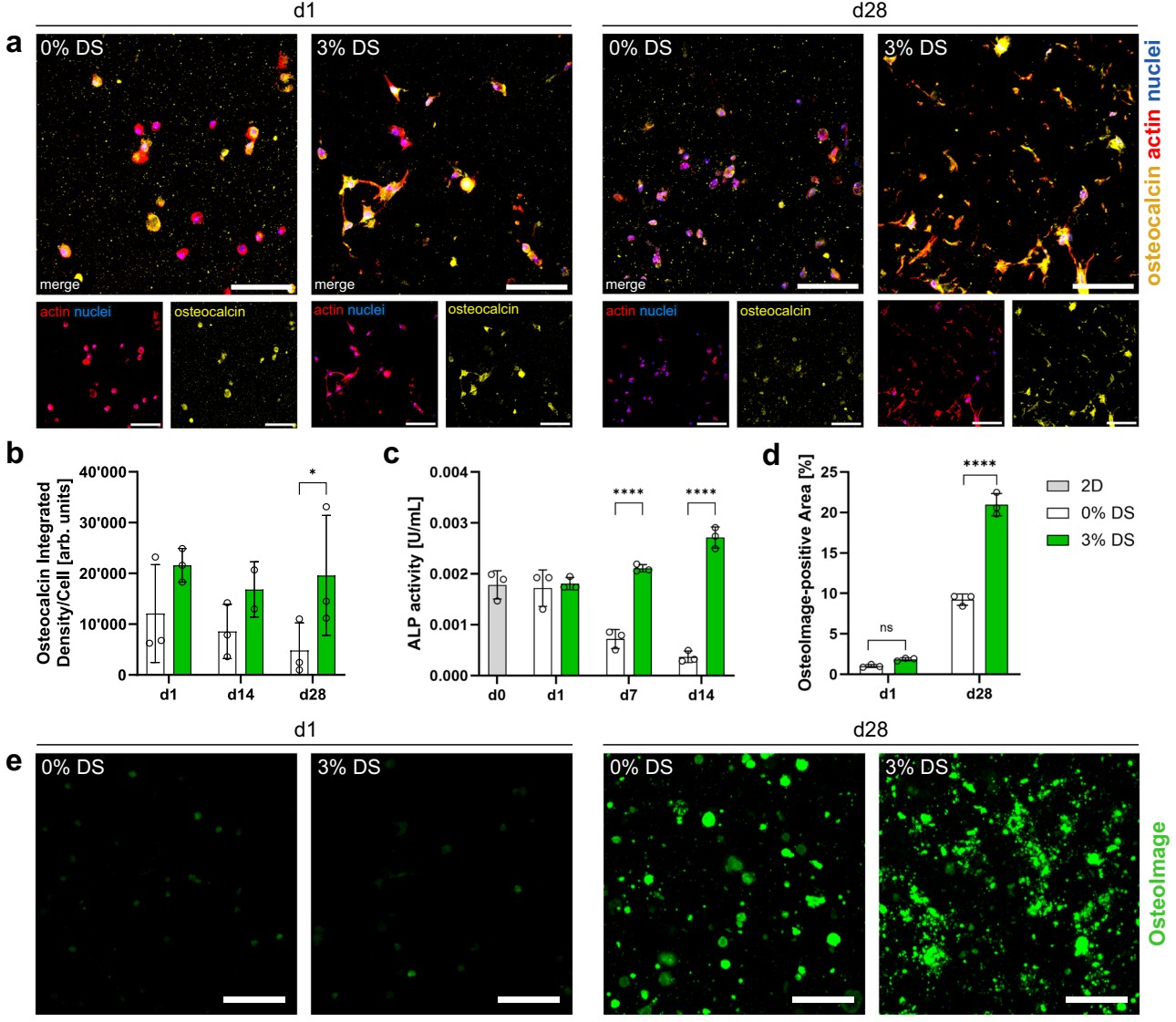

**Fig. 6 | Osteogenic differentiation of hMSCs encapsulated in microporous PVA hydrogels. a**, **b** Osteocalcin quantification in 0 and 3% DS hydrogels via immunostaining. **a** Representative maximum intensity projections of hMSCs stained for osteocalcin, actin, and nuclei at days 1 and 28. **b** The osteocalcin integrated signal density was determined from single-stain samples without phalloidin due to some spectral overlap (Supplementary Fig. 11a) and is normalized by the nuclei count ($N = 3$ samples, two-way ANOVA/Tukey). *$p = 0.0362$. **c** ALP activity in cellular samples quantified on day 0 (before 3D embedding), 1, 7, and 14 of osteogenic culture by a colorimetric assay ($N = 3$ samples, two-way ANOVA/Tukey). ****$p < 0.0001$ **d**, **e** Matrix mineralization quantified using the OsteoImage staining. **d** Percentage of OsteoImage-positive area in 0 and 3% DS hydrogels quantified from confocal images ($N = 3$ samples, two-way ANOVA/Šídák). ****$p < 0.0001$. **e** Representative maximum intensity projections of OsteoImage-stained samples on days 1 and 28. Data presented as mean ± SD. Scale bars: 100 μm (**a**, **e**).

hydrogels significantly promote HDF spreading and cytoskeletal reorganization, particularly when functionalized with the RGD adhesion peptide.

Many types of cells are known to secrete their own extracellular matrix (ECM) proteins, such as collagen and fibronectin, when cultivated in permissive hydrogels[36,37]. We thus assessed if HDFs can secrete collagen 1 within the PIPS hydrogels, and, if so, if we could accelerate its secretion by combining macromolecular crowding culture with the microporous structure. After 1 week of culture, the PIPS group demonstrated significantly higher levels of collagen 1 secretion, with both extracellular and intracellular presence (Fig. 7e, f and Supplementary Fig. 13). In contrast, the control group showed reduced collagen 1 secretion merely within the cells. To accelerate collagen secretion, the cell-laden hydrogels were cultured in a crowding medium (CM), which contains a macromolecular crowder (DS). This medium has been used to promote ECM secretion by creating a dense environment similar to natural ECM[38]. After 1 week of culture in the CM, HDFs showed a substantial increase in collagen I production compared to the two control groups. These results indicate the synergistic effect of macromolecular crowding and pore space in PIPS hydrogels to boost collagen I deposition.

## Volumetric printing with an optimized resin

Volumetric printing (VP)[18,19,39,40] is an enabling technique which allows rapid construction of 3D living hydrogel constructs in a rotating glass vial in a single step, addressing the limitations of conventional layer-by-layer manufacturing. The PIPS resins were combined with the Readily3D tomographic volumetric printer (Fig. 8a). Initial printing attempts were unsuccessful due to insufficient viscosity of the resins. In VP, a viscous or physically crosslinked resin is often needed to eliminate or limit sedimentation of structures during printing. To meet this requirement, we added 5% sacrificial gelatin to the resin as a

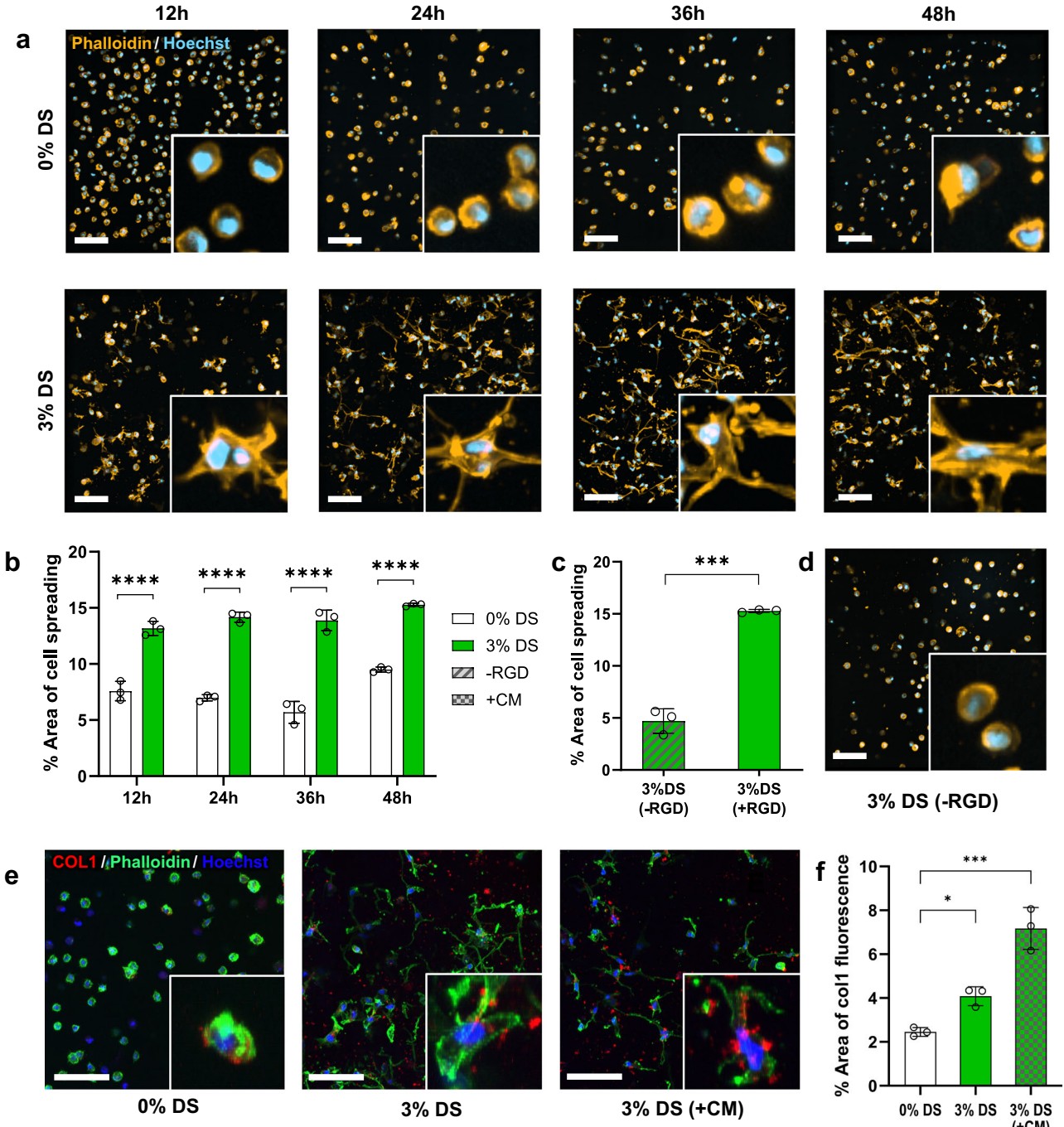

**Fig. 7 | 3D culture of HDFs in microporous PVA hydrogels. a** Representative confocal images of actin-nuclei-stained HDFs in control and PIPS hydrogels at 12, 24, 36, and 48 h. Scale bars = 100 μm. **b** Percentage of cell area in control and PIPS hydrogels (100-μm thick z-stacks, $N = 3$ samples, two-way ANOVA/Tukey). ****$p < 0.0001$. **c, d** Impact of RGD on cell spreading: quantification of cell area (**c**) and representative cell morphologies (**d**) in PIPS hydrogels without RGD peptide at 48 h ($N = 3$ samples, unpaired $t$-test, two-tailed). ***$p = 0.0001$. **e, f** Collagen secretion in different hydrogels: control (0% DS), PIPS (3% DS), and PIPS with a crowding medium (CM). Confocal images of Col1-actin-nuclei-stained HDFs after 1-week culture (**e**). Percentage of Col1 positive area (**f**) in different hydrogels ($N = 3$ samples, one-way ANOVA/Dunnett). *$p = 0.031$; ***$p = 0.0002$. Data presented as mean ± SD. Scale bars: 100 μm.

viscosity enhancer. A resin containing 2% nPVA, 1% DS, and 5% gelatin (Mix 3, Table 2) showed homogenous distribution of porosity. Unwanted phase separation was seen with 1.5% DS, while no phase separation was seen with 0.5% DS (Supplementary Fig. 14). With 1% DS, pores in the range of about 2–5 μm were observed (Supplementary Fig. 15).

Further, the resin exhibited temperature-dependent gelation. A temperature sweep from 37 to 8 °C showed a gelation point at around 20–22 °C (Supplementary Fig. 16a). Above 22 °C, the complex viscosity was about 100 mPa s (Supplementary Fig. 16b), which is almost seven times higher than the composition without gelatin. Crosslinking of the composition at 25 °C is indicated by a clear increase in G′ and G″ after UV irradiation (Supplementary Fig. 16c) until reaching a plateau in G′ of about 2 kPa within <1 min. Laser dose tests were performed to determine a suitable laser dose range for the resin. A test with small intervals was performed, and a threshold of 75 mJ cm⁻² was found

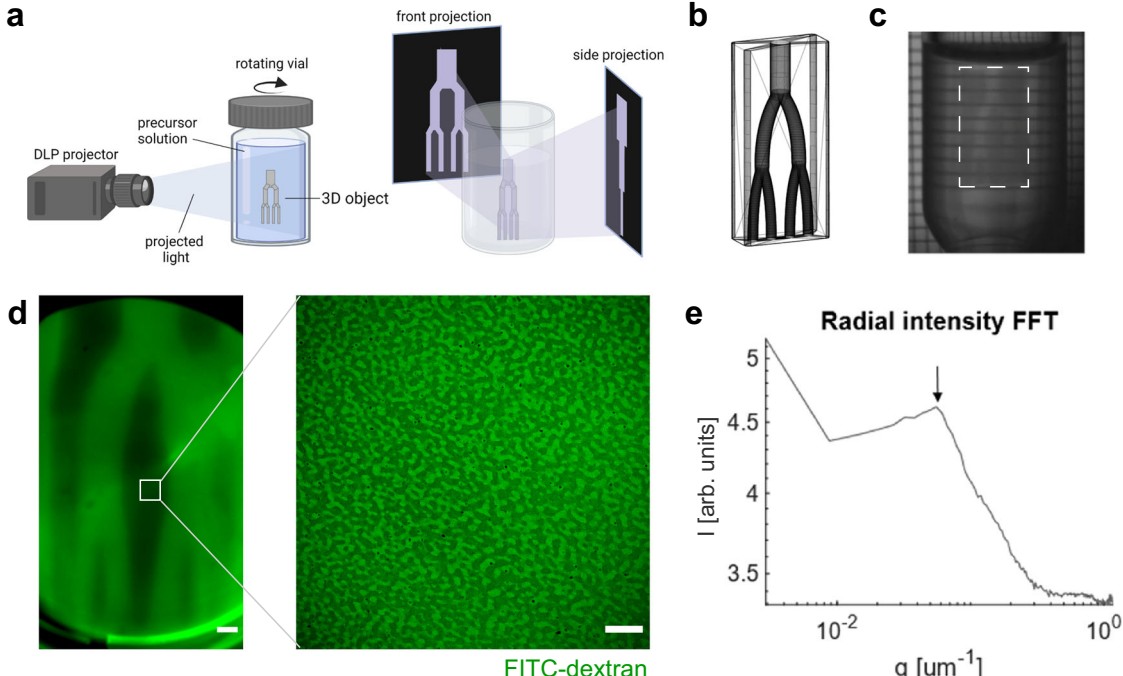

**Fig. 8 | Tomographic volumetric printing of microporous hydrogels using an optimized PIPS resin. a** Schematic representation of the volumetric printing process. Created in BioRender. Bernero, M. (2025) https://BioRender.com/avwpf01. **b** STL model of a vascularized branch model. **c** Photograph of the sample during the volumetric printing process with the appearing 3D object marked by a rectangle. **d** Representative confocal image at the center of the construct after two days in FITC-dextran. **e** RIFFT of the image shown in (**d**, right) showing a clear peak (arrow) indicating a uniform length scale of 14 μm for the porosity. Scale bars: 1 mm (**d**, left) and 50 μm (**d**, right).

(Supplementary Table 1). 3D objects could be printed with a laser dose in the range of 95 to 125 mJ cm$^{-2}$.

Using VP, a vascular branch model was printed at a laser dose of 95 mJ cm$^{-2}$ within 12 s (Supplementary Movie 4). The computer-aided design (CAD) model (Fig. 8b) has a height of 1.4 cm, and vascular channels with dimensions of 0.7–1.4 mm. The turbid construct is depicted while printing (Fig. 8c). Channels were shown to be perfusable after leaching out the gelatin for 2 days at 37 °C and subsequent incubation in a FITC-dextran solution for one day (Fig. 8d). Next, we investigated if the printed constructs are indeed microporous. After incubation in PBS and FITC-dextran, confocal imaging data showed interconnected pores in the printed gel constructs (Fig. 8d). The RIFFT of this image showed a clear peak corresponding to a characteristic length scale of around 18 μm (Fig. 8e). A proof-of-concept experiment showed that the branch structure could be printed in the presence of live hMSCs. After bioprinting, the channels were perfused by a phenol red-stained medium (Supplementary Fig. 17a). The gel appeared to be microporous (Supplementary Fig. 17b).

## Conclusion

In conclusion, we developed a class of microporous hydrogels by PIPS with cell-guiding properties for 3D tissue culture. The pore size could be fine-tuned by the gel composition as well as light intensity. Compared to other hydrogels relying on polymerization-induced phase separation[12,13], our PVA-based PIPS hydrogel system provides the following advantages: (1) nPVA has >30 clickable ene groups and offers more flexibility for conjugating bioactive peptide motifs than 4-arm PEG precursors; and (2) it does not rely on the high viscosity of hyaluronan and thus avoids possible cell damage due to high shear stress during mixing. These hydrogels support in situ 3D photoencapsulation of living cells, high cell viability and fast cell spreading, differentiation and matrix secretion over up to 4 weeks. Integrating PIPS with VP allowed generating cm-scale microporous hydrogel constructs within

20 s. Besides VP, the PIPS hydrogels may find applications in other light-based 3D printing, such as digital light processing (DLP)[41,42], extrusion printing[43,44], and two-photon direct laser writing[22,27,45]. Using light as the trigger, different pore sizes may be digitally 3D-printed within one object by tuning the light intensity and PIPS. Accordingly, complex hierarchical structures with both macropores and micropores could be generated in the presence of living cells. Integration with perfusion bioreactors may further enhance tissue maturation. Given that the disclosed PIPS hydrogels are not proteolytically degradable, optimization with protease-sensitive motifs may enhance cell-matrix remodeling, 3D cell infiltration, and in situ tissue regeneration. This system holds the potential to significantly advance the fields of biofabrication and functional tissue engineering.

## Methods
### Materials

A final concentration of 0.05% LAP was used throughout all experiments with UV curing. nPVA (54 kDa, DoF 3.5%) was synthesized as described elsewhere[46] (see Supplementary Information). It was used at different concentrations and dissolved in a solution containing the photoinitiator by vortexing and ultrasonication. PEG-2-SH (2 kDa, LaysanBio) was used as a crosslinker with a concentration that results in a thiol/ene ratio of 1.6. It was dissolved freshly in PBS. DS (40 kDa, Carl Roth) was used for phase separation and was dissolved in PBS at 50 °C for 2 h by vortexing. Dextran (40 kDa, Sigma-Aldrich) was dissolved similarly. In cell culture experiments, RGD-functionalized nPVA was added to the resins to promote cell attachment. All components were dissolved in PBS. A detailed description of resin compositions is included in the Supplementary Information.

### Approximation of a phase diagram

We employed a combination of techniques as described elsewhere[47] to determine the approximate phase diagram, with a particular emphasis

on the cloud point (CP) method. This method relies on visual turbidity to identify the binodal, and the CP is the point at which a solution turns and remains turbid during the titration of a polymer solution with another. Vials with a stock solution of 10% nPVA were prepared and then diluted with water, providing a set of varied starting concentrations between 0.25 and 8% nPVA. These samples were individually titrated with 50% DS in small steps until they reached their CPs. Careful observation of disappearing turbid streaks helped in assessing proximity to the binodal. The DS concentration was lowered in response to appearing streaks to increase the accuracy of the titration. The last drop to achieve the CP was aimed to have a size of 0.1 µL for increased accuracy. The DS concentrations needed for transition to a turbid solution were recorded as CPs. Finally, the binodal curve was calculated in RStudio through linear-quadratic regression of the obtained data points.

### Hydrogel preparation
The components were thoroughly mixed by pipetting up and down 20 times after each additional component. With large volumes (mL), the composition was mixed until it turned visibly clear, whereas it was mixed 20 times and then vortexed for two to three seconds with small volumes (µL). The crosslinker was usually added last. A detailed description of crosslinking and casting the gels in molds can be found in the Supplementary Methods.

### Turbidity measurements
The turbidity was determined by measuring the absorption at 405 nm with a plate reader (Spark M10, Tecan). Before curing, 40 µL of the mixed precursor solution was loaded into a well of a 96-well plate. Gels cured in Teflon molds were punched with a 5 mm biopsy punch to fit into the plate, and the turbidity was measured after adding 100 µL PBS into the well ($N = 5$ samples).

### Permeability of microporous hydrogels with FITC-dextran
The permeability of gels with FITC-dextran (500 kDa, Sigma-Aldrich) was studied similarly as described elsewhere[48]. In short, the gels were washed for 1–3 days at 4 °C in PBS till reaching a swelling equilibrium. Then, PBS was replaced with a solution of FITC-dextran (1 mg/mL), and the samples were imaged after 1–2 days of incubation.

### Fluorescent labeling of hydrogels
For confocal imaging of the hydrogels before and after UV crosslinking, either 1/50th of FITC-labeled nPVA was included in the resin or methacryloxyethyl thiocarbamoyl rhodamine B (rhodamine-MA, Polysciences) was added to the hydrogel mix (0.1% stock solution, diluted 1:500). The rhodamine-MA is conjugated to the hydrogel during photocrosslinking.

### 2D Fast Fourier transform
2D-FFT is a common way to characterize the length scale of pores in bicontinuous structures[49]. The analysis was done with a MATLAB code, which can be found in the Supplementary Information. In short, an image is read and converted to greyscale. Then the 2D-FFT of the image multiplied by a Hamming window is taken. Further, radial binning is performed to show the radial intensity of the FFT (RIFFT). The length scale was then calculated as $q^{-1}$.

### 3D cell culture
Both hMSCs and HDFs were embedded at a density of $4 \times 10^6$ cells/mL within the hydrogel matrix. The cells were thoroughly mixed with the gel precursors (see Supplementary Table 3) and the mixture was subjected to UV irradiation (10 mW/cm²) for 5 min to initiate crosslinking. After curing, the cell-laden gels were washed with warm PBS for at least 5 min to remove unreacted components. The hMSC hydrogels were then cultured in osteogenic medium (DMEM, 10% fetal

bovine serum (FBS), 1% antibiotic-antimycotic, 50 µg/ml ascorbic acid, 100 nM dexamethasone, 10 mM beta-glycerophosphate) and the medium was changed three times per week. The HDF-laden hydrogels were then cultured in a complete growth medium (DMEM, 10% FBS, 1% antibiotic-antimycotic) for subsequent experiments. To assess the ability of the PIPS hydrogels to support collagen production, the HDF-laden gels were also cultured in a crowding medium containing DMEM, 0.5% FBS, 100 µM L-ascorbic acid, and 100 µg/mL 500 kDa dextran sulfate.

### Statistical analysis and reproducibility
Statistical analysis was performed in GraphPad Prism 8.2.0, except for the slope analysis, which was performed in Excel. Ordinary one-way or two-way analysis of variance (ANOVA) were performed depending on the number of variables. They were followed by Šídák's or Tukey's test for multiple comparisons for two or more groups, respectively. If the standard deviations did not appear equal among groups, the Brown–Forsythe and Welch ANOVA test, followed by Dunnett's T3 multiple comparison test, was applied. In the two-way ANOVA, only simple effects within rows and columns are analyzed. Welch's $t$-test was used when there were only two groups. $P$ values less than 0.05 were considered significant. The outcomes obtained from acellular experiments and 3D hydrogel culture were consistent and replicable in at least two independent experiments, with a minimum of three replicates in each experiment. An exception was the proof-of-concept osteogenic culture of 28 days, performed once.

### Reporting summary
Further information on research design is available in the Nature Portfolio Reporting Summary linked to this article.

## Data availability
The data that support the findings of this study are available in the ETH Zurich Research Collection with the identifier [DOI: 10.3929/ethz-b-000734397]. Data is available from the corresponding author on request.

## Code availability
The code used for the RIFFT analysis can be found in the Supplementary Information. Scripts are available from the corresponding author on request.

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

## Acknowledgements

This work was supported by the Swiss National Science Foundation (nos. 190345, 188522, and 206501) and the Swiss Secretariat for Education Research and Innovation (no. MB23.00008). M.Z.M. acknowledges the financial support from the ETH-D Excellence Scholarship program. M.B.

likes to acknowledge support from the ALIVE | Advanced Engineering with Living Materials initiative of ETH Zurich, which is funded by the ETH domain Strategic Focus Area Advanced Manufacturing program. W.Q. gratefully acknowledges financial support from the China Scholarship Council (no. 202006790027). We thank Martin Ehrbar and Alain Plüss for scientific discussion. We thank Martina Civico Ramos for experimental support, Doris Zauchner for her help with data processing, Paul Delrot (Readily3D) for technical assistance, and Alba Sicher for providing the MATLAB code. We also gratefully acknowledge staff members at the ETH Zurich ScopeM for their support and assistance in this work.

## Author contributions

Conceptualization: X.-H.Q. and M.Z.M.; Experiments and visualization: M.Z.M., M.B., C.X., W.Q., E.O., L.R. and X.-H.Q.; Data analysis: M.Z.M., M.B., C.X., W.Q., E.O., T.C.T.M., R.W.S. and X.-H.Q.; Methodologies: M.Z.M., M.B., W.Q., E.O. and X.-H.Q.; Writing of first draft: M.Z.M. and X.-H.Q.; Reviewing of final manuscript: M.Z.M., M.B., C.X., W.Q., E.O., L.R., T.C.T.M., R.W.S., R.M. and X.-H.Q.

## Competing interests

ETH Zurich (with X.-H.Q., M.Z.M., R.M. and W.Q.) has filed patent applications for this in vitro cell culture technology. The inventors declare no other competing interests. Correspondence and requests for materials should be addressed to X.-H.Q. Additionally, it is confirmed that no other authors involved have any competing interests.
