## [Transparent Peer Review file · Nature Communications]

Cell-guiding microporous hydrogels by photopolymerization-induced phase separation

Corresponding Author: Professor Xiao-Hua Qin

Version 0:

Reviewer comments:

Reviewer #1

(Remarks to the Author)

In this manuscript, the authors introduced a photocurable resin to fabricate cell-laden macroporous hydrogels for better solute transport and cell-cell communication. The porosity in hydrogels is formed via photopolymerization-induced phase separation (PIPS) using two ionic polymers. Though the PIPS mechanism has been well investigated in existing literature, the current manuscript presents a comprehensive optimization process based on nPVA and dextran sulfate. Regarding biological outcomes, more evidence might be needed to support the commitment of cell-cell communication. Please check out my comments below:

- 1- Phase separation is a powerful approach to introducing micropores to hydrogels. Previous studies by Yu Zhang et al. have reported the use of aqueous two-phase systems to engineer porous bioinks and thus porous hydrogel structures fabricated via bioprinting (including light-polymerization-based approaches). The authors should touch more on such relevant work in the introduction and make proper comparisons. For example, what are the advantages of their approach over existing ones? In other words, what drives the authors to perform such a study? I feel the porosity and cell culture outcomes can also be achieved by existing approaches. The authors need to demonstrate the novelty and significance of their work and clarify the new contribution.
- 2- The cells encapsulated in the hydrogels do not seem to proliferate (figure 5C, 7e). I am wondering why the MSCs (a cell type supposed to grow fast) wouldn't proliferate in the materials. Without supporting the growth of cells, such a resin system might be less useful for cell culture and tissue engineering.
- 3- The current cell density in the resin seems quite low ($0.5 \times 10^6/\text{ml}$). Why do they choose such a low concentration? Is it possible to include more cells while still maintaining the porosity formation?
- 4- The cell viability of day 1 is presented. I suggest including the viability data throughout the culturing process. This might provide some clues on why the cells do not proliferate.
- 5- In Figure S10 "Pores are penetrated by cell processes", two cells are presented. One is quite elongated and the other one is rather round. Does this mean that the cell behavior is randomly affected by the porosity?
- 6- The authors found a narrow window of the polymer concentrations to achieve PIPS. Would this hinder the use of their system in different applications (e.g., culturing different cells).
- 7- Does the term macroporous hydrogel indicate the one with pores of 2-40 μm ? Would microporous hydrogel be more appropriate?
- 8- Figure 1c indicates a hydrogel with phase separation. Why is it clear?
- 9- what causes the delay of photocrosslinking? Tens of seconds of delay are likely to affect the fabrication process. Isn't it?
- 10- Figure 2d. After swelling, the nPVA phase (crosslinked domain) seems to be much smaller. Why?
- 11- In Figure 2c, the authors mentioned the formation of droplets for crosslinked nPVA. Any 3D confocal reconstruction to show this? If crosslinked droplets were formed, after removing dextran phase, the structure should collapse. Why would the connected pores be seen after swelling in Figure 2D?
- 12- Figure S3a indicates many micropores to me. Why would the authors say that no pores were formed?
- 13- In Figure 2c, both groups contain the same amount of FITC-nPVA before UV. the fluorescence seems to be darker with the 2.5% DS group. why?
- 14- The authors incubated the porous hydrogel in FITC-dextran for 1-2 days to locate the pores, which seem to be long. Is this duration of incubation necessary? Is the pores are well connected, the penetration of dextran should be fast. Would be helpful to characterize the connectivity of the pores.

15- The porosity and pores size should also be characterized under different concentrations since the pores are the key point of the work.

Reviewer #2

(Remarks to the Author)

The authors report a new hydrogel system based on modified polyvinyl alcohol, dithiol crosslinkers, a water-soluble photoinitiator and dextran sulfate for bioprinting 3D structures with controlled macroporosity. Macroporosity in the printed structures is enabled by photopolymerization-induced phase separation during the printing process and is controlled by means of the light intensity and the content of modified dextran in the system. The material is characterized in terms of its porosity, mechanical properties and turbidity as a function of the concentration of its components and the light intensity used for photocrosslinking. The authors also show that it is possible to use the material for volumetric bioprinting of cell-embedded hydrogel structures.

The paper introduces a PIPS process compatible with live cells and volumetric bioprinting. However, it is not very clear from the paper how this process could significantly improve bioprinting applications. In addition, different materials and different degrees of functionalization of nPVA have been used at different stages of the project, which makes it difficult to compare the characterization results. For example, the mechanical properties of the hydrogels were characterized using hydrogels not containing gelatin, a cell-adhesive peptide, pyrogallol and iodixanol, which were all added to the bioresin used for the bioprinting experiments. In particular, the influence of gelatin (as a sacrificial viscosity enhancer) in the PIPS process and in the macroporosity of the resulting 3D printed matrices should be investigated and shown. Moreover, an experiment that shows how PIPS improves the biological outcome of volumetric bioprinting or that shows a clear application which would not be possible without PIPS process would be highly valuable to improve the manuscript and show its contribution to the field of biofabrication.

Therefore, I cannot accept this paper for publication in NC. Here are some more specific comments:

1. In the abstract, the reported hydrogel system is claimed to be "low-cost" (line 36, page 2). Such claims should be avoided when proper evidence is not provided. Proper evidence includes a cost analysis of acquiring the materials and processing them into the precursors of the system and a cost comparison with other potentially similar hydrogel systems available on the market.
2. From the introduction, it is not clear what the potential outlook of this work is. What is being enabled that was not possible before? How is this work specifically contributing to move forward the field of biofabrication?
3. I suggest the authors include a crosslinking or printing window diagram. This will be helpful to (i) complement the phase diagram and (ii) to visually show the range of concentrations for each component of the hydrogel system where PIPS can happen and a porous crosslinked hydrogel can be obtained. This should be also done for the compositions used for volumetric printing, which were clearly different from the ones used for characterization. Some examples of this can be found in "Naomi Paxton et al 2017 Biofabrication 9 044107".
4. The purpose of fitting a curve to the pore size vs. light intensity data and providing the equation with the best fit is not clear. Moreover, goodness-of-fit measures such as the standard error of the regression are not provided. The purpose of including this curve fitting in the manuscript should be clearly stated and goodness-of-fit measures should be included along with the equation. Otherwise, such data should not be included in the manuscript.
5. According to the manuscript, gelatin was added just to enhance the viscosity of the bioresin composition and is not expected to participate in the crosslinking reactions, thereby acting as a sacrificial material. The concentration of gelatin added to the system is high when compared to the main polymer to be crosslinked, which is nPVA (gelatin 5 % vs. nPVA 2 %). How does the presence of gelatin during the photocrosslinking process influence the pore formation and the properties of the final matrix? What proportion of the initially added cells to the bioresin are being washed away with the gelatin from the bioprinted construct when incubating it at 37 °C in cell culture medium? This might be related to the low cell density evidenced in Figs 7 and S13.
6. In Table 1, the mechanical properties reported for the hydrogel composition including gelatin (mix 3) were measured before washing out the gelatin, therefore not reflecting the mechanical properties of the final 3D printed construct. The reported values cannot be related to the mechanical properties of the matrix where cells are being cultured for two weeks after the bioprinting process.
7. Figure S16. The printed structure is not clearly visible. Is it a monolith-like structure? The images from d14 show stronger red staining in the samples without iodixanol. Does this suggest that mineralization decreases when using iodixanol? Can this observation be attributed to some effect of the iodixanol on the bioactivity of the bioresin or to its inherent effect on the optical properties of the material? A hypothesis for this phenomenon should be presented and hopefully investigated. Further phenotypic characterization of the cells cultured within the printed construct (with PIPS vs. without PIPS) might shed light on the possible advantages of the apparent macroporosity conferred to the matrix by PIPS.

Reviewer #3

(Remarks to the Author)

Reviewer #4

(Remarks to the Author)

The manuscript entitled, 'Photoclick Phase-separating Hydrogels for 3D Cell Culture and Volumetric Bioprinting' reported the super-fast volumetric bioprinting of cm-scale macroporous hydrogels with encapsulated 3D hMSC. This article is interesting, but there are major gaps. In its current state, it is not up to the standard of Nature Comm.

Specific comments:

1. Direct fabrication of cell-laden hydrogels using photo-initiated crosslinking of polymers, such as Gelatin Methacryloyl (GelMA) and Polyethylene Glycol Diacrylate (PEGDA), has been reported extensively over 2 decades. The pore size and porosity of these hydrogels are known to support cell growth. The claim in the introduction, "Existing hydrogel systems ... have very small pore sizes (5 – 50 nm) and fail to provide a permissive environment for encapsulated cells" is wrong. The authors should provide the pros and cons of the reported system in comparison to the existing hydrogel systems for 3D cell culture.
2. Hydrogel matrices for 3D cell culture must be biodegradable in order to be replaced by the natural ECM produced by the encapsulated cells and to make room for cell proliferation. No biodegradability data is provided in this report.
3. Stem cells will only proliferate and differentiate properly in an ECM that mimicks the stem cell niche. The reported system lacks cell binding sites and other biochemical signatures for cell-matrix interaction. As shown in the data, hMSC proliferation is very slow. There is no hMSC phenotype data.
4. The authors reported about the alkene functionalization of PVA. The degree of alkene substitution is not clear. Even though the synthesis method has been reported previously, characterization data for the nPVA used in this study should still be provided.
5. Thiol-ene chemistry is sensitive to light and air, which could lead to premature degradation. How has this been taken into consideration?
6. The authors mentioned bimodal behavior of the polymer premix. Did they check the turbidity point?
7. The authors stated that '...phase-separating resin was compared to a composition with the nonionic dextran instead of DS...'; but there are several reports on the dextran based macroporous gels. Some references are needed to justify this.
8. There is no change in water uptake with alkene content, why? In general, degree of gelation is directly related to the crosslinking density and water uptake.
9. How was the porosity measured? The pore size distribution is not provided either.
10. What does 'higher crosslinking density in the nPVA-rich phase' mean? By definition, a hydrogel consists of both solid and liquid phases. The solid phase is crosslinked. Are the authors suggesting that within the "solid" phase of the reported system, there are more than one phases? If so, what are they? And what are the estimated crosslinking densities?

Reviewer #5

(Remarks to the Author)

Version 1:

Reviewer comments:

Reviewer #1

(Remarks to the Author)

The authors have included new experiments and data to improve the manuscript. I appreciate the authors' effort. However, some of my previous concerns were not fully addressed.

1-Novelty and new contribution (following my 1st comment in the previous round).

- a. Other phase separation approaches have already proven the benefits of micropores for cell growth. The current manuscript draws the same conclusion using the so-called PIPS. Again, the new contribution is not clear. Is it better than existing ones? Why? I think the other two reviewers also expressed similar concern as the results look so similar to existing literature. The intensively cited reference (Ying et al. Adv Mater 2018) is just an early example. Recently, more and more aqueous two-phase or coacervation microporous hydrogel systems (<https://doi.org/10.1002/adhm.202203243>; <https://doi.org/10.1002/adfm.202400431>; <https://doi.org/10.1002/adma.202300636>; <https://doi.org/10.1002/adma.202410452>) have been reported. The recent work from Tibbitt et al. published in Advanced Materials (<https://doi.org/10.1002/adma.202410452>) actually reported the polymerization-induced phase separation (PIPS) system to introduce pores to hydrogel for 3D cell culture. The fibroblasts (also used in this manuscript) look much more spread than those in this submission. Together, though the authors have included more experiments and data, the core novelty and new contribution of the study are still lacking. The publication of similar PIPS work in other journals further declines the novelty.
- b. In the response, the authors explained the differences between aqueous two-phase and PIPS approaches. They stated that the PIPS approach allows for in situ pore formation for 4D tissue and organoid engineering. This statement is not evidenced. If the authors can really identify some important features for cell culture that the aqueous two-phase system cannot deliver, their work would really make a difference.

2-following my 4th comment in the previous round.

The authors claimed that they did not notice any cell number reduction over the 28 days of culture. Looking at the nuclei (blue DAPI dots) (Figure S9), I think one can see fewer cells on day 28 than on day 3. The authors should double-check

such details to avoid misleading.

3-Photocrosslinking mechanism (following my 9th comment in the previous round)

a. In the original and revised figures (e.g., figure 4c), there is a considerable delay in the increase of modulus in the presence of UV light. In the first tens of seconds, the hydrogel didn't seem to crosslink. what was happening then? Would phase separation happen during this process?

b. Was the delay caused by the PIPS system?

c. My previous concern was that the delay might affect the biofabrication process. Volumetric bioprinting claimed super fast fabrication (down to tens of seconds). If the PIPS will only crosslink after tens of seconds of UV irradiation, how to incorporate it with such biofabrication settings?

Reviewer #2

(Remarks to the Author)

The authors have significantly modified their manuscript taking into account reviewers suggestions and focused more on the material itself and cell culturing and less on 3d bioprinting. I think it was the right decision. The formation of "microporous" hydrogel in-situ in the presence of cells by PIPS is an interesting and important approach, especially for bio applications. Mechanical properties of different materials have been thoroughly investigated and characterized. The effect of porosity on cells has been also investigated and there is indeed a significant positive effect.

Overall, I am willing to recommend the new version of the paper for publication with minor revision. Below are a few minor suggestions for improvement.

One question/suggestion. The field of porous materials has a problem of definitions. Traditionally people separate them into micro- (below 2 nm), meso- (2-50 nm) and macroporous (above 50 nm). This however is not the most convenient nomenclature in my opinion. Nevertheless, I recommend the authors to add one sentence giving their definition of "microporous" materials to avoid possible confusions.

Scale bars (values) are not visible in some images.

Reviewer #3

(Remarks to the Author)

Reviewer #4

(Remarks to the Author)

The authors only addressed the reviewer comments partially. Overall, there are not enough new concepts or sufficient novelty to warrant publication in Nature Communication. Even though the authors provided additional data to support porosity and cell-compatibility, they failed to answer the very fundamental question: how is their system/composition significantly better than the existing hydrogel-based scaffolds? Hydrogel, by definition, consists of two phases: solid and water. The use of photopolymerization to produce hydrogels has been reported extensively, including many cell-laden systems with interconnected macropores.

Reviewer #5

(Remarks to the Author)

Version 2:

Reviewer comments:

Reviewer #1

(Remarks to the Author)

After looking into the revised manuscript and the authors' response (including those to other reviewers), I still hold my point that the work lacks sufficient impact and novelty to be published in Nature Communications.

1-Regarding the novelty and new contribution, I still don't think the current manuscript leads to considerable new knowledge

or new approaches.

a. The authors acknowledge the existing competing studies but only include one that was published before their submission date. However, they did cite some newer literature during revision in discussions other than the core novelty session. I think they should make the rule consistent and include the most relevant references, especially when communicating their novelty and contribution.

b. The authors didn't provide evidence but just repeated their statements regarding the 4D tissue and organoid engineering. They don't even intend to change their tone with the claim. This point is associated with the biological outcomes of their material system. Again, they failed to prove the advances of their system over existing phase separation systems regarding cell activities and biofunctionality.

2-The authors seem to realize the conflict between their claim and figure S9, and thus have changed the text. This could be one example, the authors should carefully check the manuscript to make sure they have properly describe their results. In response Figure 1, more cells were seen in the DS-free control group than the PIPS group at day 1 and day 7. At day 1, the cell number of 3% DS group is just 1/3 of that of 0% DS group. Why? Shouldn't they fix the initial cell amount? does this mean that the PIPS will lose a large number of cells due to the removal of the sacrificial phase? If yes, this is a major drawback.

3-the delay of photocrosslinking is not clarified properly. They evidenced that the delay was not caused by the PIPS system. But again, how could bioprinting happen within 12s when the first tens of seconds of light irradiation do not induce polymerization? If the addition of gelatin will enhance the photoreactivity, the authors should present the photorheology of the formulations containing gelatin.

Reviewer #2

(Remarks to the Author)

The authors have satisfactorily addressed all the reviewers' comments and the paper can be published after minor revision. The use of in-situ PIPS to create hierarchically porous hydrogels that enhance cell behavior in 3D cell culture is an important concept, which the authors convincingly demonstrate. The observed decrease in cell number after 28 days of culture is not surprising, but it should be further investigated in future studies using specific cell systems, as this effect is likely to vary depending on the cell type used. The delayed increase in modulus may result from the interplay between phase separation (which is kinetically slower than polymerization) and polymerization. Regarding the paper by Tibbitt and colleagues, it should be properly cited in the manuscript, and any differences or similarities should be clearly and transparently discussed to ensure fairness to the research community.

REVIEWER COMMENTS

Reviewer #1 (Remarks to the Author): In this manuscript, the authors introduced a photocurable resin to fabricate cell-laden macroporous hydrogels for better solute transport and cell-cell communication. The porosity in hydrogels is formed via photopolymerization-induced phase separation (PIPS) using two ionic polymers. Though the PIPS mechanism has been well investigated in existing literature, the current manuscript presents a comprehensive optimization process based on nPVA and dextran sulfate. Regarding biological outcomes, more evidence might be needed to support the commitment of cell-cell communication. Please check out my comments below:

Authors' response: We are grateful for the reviewer's positive feedback and constructive suggestions. We have performed new cell culture experiments to strengthen the biological outcomes of this manuscript. Our findings confirm the commitment of cell-cell communication within the microporous hydrogels by PIPS.

1- Phase separation is a powerful approach to introducing micropores to hydrogels. Previous studies by Yu Zhang et al. have reported the use of aqueous two-phase systems to engineer porous biinks and thus porous hydrogel structures fabricated via bioprinting (including light-polymerization-based approaches). The authors should touch more on such relevant work in the introduction and make proper comparisons. For example, what are the advantages of their approach over existing ones? In other words, what drives the authors to perform such a study? I feel the porosity and cell culture outcomes can also be achieved by existing approaches. The authors need to demonstrate the novelty and significance of their work and clarify the new contribution.

Authors' response: Thank you for this constructive comment. We are aware of the seminal work of Zhang lab in applying aqueous two-phase systems to engineer porous hydrogels (Ying et al. Adv. Mater. 2018). However, it significantly differs from our approach because: in former case, before light exposure the resin mixture has formed an emulsion consisting of GelMA and PEG phases (which are then arrested by photopolymerization), whilst our formulation remains miscible (one-phase) before light irradiation and it only phase separates into porous structures upon light exposure as a result of increased M_w and resulting incompatibility of the mixture. Importantly, our PIPS approach uniquely allows in situ pore formation induced by light in the presence of living cells. This is an important advance because light-induced in situ pore formation provides a tool for 4D tissue and organoid engineering.

To address this comment, we have revised the manuscript as follows:

“Ying et al.⁹ reported aqueous two-phase systems to engineer microporous polyethylene glycol (PEG)-gelatin methacryloyl hydrogels. Photopolymerization is applied to arrest two distinct phases in an emulsion.”

2- The cells encapsulated in the hydrogels do not seem to proliferate (figure 5C, 7e). I am wondering why the MSCs (a cell type supposed to grow fast) wouldn't proliferate in the materials. Without supporting the growth of cells, such a resin system might be less useful for cell culture and tissue engineering.

Authors' response: We have performed new 3D cell culture experiments. The results demonstrate that our microporous hydrogels support the growth of both human fibroblasts and human mesenchymal stromal cells (hMSCs) as well as osteogenic differentiation of hMSCs.

Although hMSCs proliferate when cultured in an expansion medium, it should be noted that we switched to an osteogenic culture medium after the cells were embedded. hMSCs cultured under osteogenic conditions are expected to shift to a differentiation state with reduced proliferation. (Zhang et al. Acta Biomaterialia 2022)

(Source: Figure 3, Zhang et al. 2022 Acta Biomaterialia)

Therefore, we focus on quantification of cell viability, which is very high (>98%) in the PIPS hydrogels over 7 days (See **Revised Figure 5a-b**).

Revised Figure 5a-b

3- The current cell density in the resin seems quite low ($0.5 \cdot 10^6/ml$). Why do they choose such a low concentration? Is it possible to include more cells while still maintaining the porosity formation?

Authors' response: We used the low cell concentration due to practical constraints. But this resin does allow 3D photoencapsulation of cells at a much higher density (5 Mio per ml) while allowing *in situ* pore formation by PIPS. We have successfully performed new 3D cell culture experiments using both human fibroblasts and hMSCs at a density of 4 Mio/ml (see **Revised Figures 5-7**).

Additionally, the pore-forming characteristics of the cellular gels were also analyzed to confirm successful PIPS (see **Revised Supplementary Figure 9**).

4- The cell viability of day 1 is presented. I suggest including the viability data throughout the culturing process. This might provide some clues on why the cells do not proliferate.

Authors' response: We have included the new cell viability data on days 1 and 7 showing a consistently high viability in the PIPS group (see **Revised Figure 5a-b**). We have not noticed any cell number reduction over the remaining culture time up to day 28 (see new **Supplementary Figure 9**). As also discussed in reviewer question 1.2, we are not concerned about slow proliferation, since we

anticipate the cells to redirect towards the osteogenic differentiation pathway as described in our previous work (Gehlen et al. Acta Biomaterialia 2023; Zhang et al. Acta Biomaterialia 2022).

5- In Figure S10 "Pores are penetrated by cell processes", two cells are presented. One is quite elongated and the other one is rather round. Does this mean that the cell behavior is randomly affected by the porosity?

Authors' response: Thank you for pointing this out. We performed new cell cultures at higher cell densities and optimized the method of incorporating the cell-adhesive RGD peptides. Instead of adding soluble cysteine-containing RGD peptides, we have instead first conjugated RGD to nPVA to ensure that the peptides are firmly immobilized on the hydrogel formed. The results are shown in **Revised Figure 5d** and highlight the cell processes within the pores, additionally an animation of the entire stack can be seen in the new **Supplementary Movie 3**. With the increased cell density and optimized RGD-functionalization, we have demonstrated a consistent effect on 3D cell morphogenesis within the microporous hydrogels, which can also be seen over time in the new **Supplementary Figure 9b**.

Revised Figure 5d

Revised Figure S9b

6- The authors found a narrow window of the polymer concentrations to achieve PIPS. Would this hinder the use of their system in different applications (e.g., culturing different cells).

Authors' response: Thank you for your comment. The objective of this work is to apply PIPS in 3D cell culture applications and proof-of-concept volumetric printing. We agree with the reviewer that selection of the appropriate compositions is key to PIPS. As explained in **Revised Figure 2d-e**, we defined measures for PIPS and determined the window for PIPS. The results show that the polymer concentrations cover a wide range (1-5%). This manuscript focused on equal amount of polymer concentration of nPVA and DS in 2-3 % to ensure the mixture remains 1 phase before photopolymerization.

Revised Figure 2d-e

We have revised the manuscript as follows (Section 2.2, Page 4):

“Next, we experimentally screened the feasibility of different resin compositions for PIPS on an optical microscope (Fig. 2d-e). Phase-contrast images were taken before and after UV curing. When both nPVA and DS are below 2.5%, no phase separation is observed before nor after UV. At higher concentrations, PIPS occurred as evidenced by the transition from an optically clear mixture (before UV) to two phases (after UV). However, further increase of either nPVA or DS resulted in emulsions where the compositions demixed before UV.”

7- Does the term macroporous hydrogel indicate the one with pores of 2-40um? Would microporous hydrogel be more appropriate?

Authors' response: Thank you for the suggestion. We have revised the title and mentions to “microporous”.

8- Figure 1c indicates a hydrogel with phase separation. Why is it clear?

Authors' response: As our hydrogels undergo PIPS, they turn from fully transparent to a turbid state which can be seen with the dark background in **Figure 1c**. We have now also included a

comparison between a non-PIPS hydrogel composition and a PIPS hydrogel in **Revised Figure 2c**, where the turbidity after PIPS can be seen.

Revised Figure 2c

9- what causes the delay of photocrosslinking? Tens of seconds of delay are likely to affect the fabrication process. Isn't it?

Authors' response: The delay is because of low light intensity applied. Depending on the light intensity, the crosslinking dynamics are different, which influences phase separation. Accordingly, the resulting pore morphology of the gels can be tuned by the light intensity via the crosslinking dynamics.

10- Figure 2d. After swelling, the nPVA phase (crosslinked domain) seems to be much smaller. Why?

Authors' response: The images before and after swelling in **Figure 2d** were not acquired from the same samples because of the spectral overlap between the FITC-labeled nPVA and the FITC-dextran tracers used for perfusion. We thus repeated the experiment with rhodamine-labelled hydrogels and acquired images immediately after UV crosslinking as well as after two days of swelling and FITC-dextran incubation. The results suggest that there is no clear swelling trend and the characteristic pore length scale remains similar. An exception is the slightly increasing pore size in 3% nPVA-2.75% DS hydrogels, which, apart from swelling, might also be due to washing out of unbound rhodamine dye (note decrease in total fluorescence intensity and clearer pores).

Response Figure 1

We have thus revised **Figure 2f** and the text, Section 2.2, Page 5, as follows:

“Since the pore size may change dramatically after swelling, imaging was repeated after 2 days. The pore length scale for the 3% DS group did not change significantly, although small changes were evidenced in another composition (Supplementary Fig. 2b-c)”

Revised Figure 2f

11- In Figure 2c, the authors mentioned the formation of droplets for crosslinked nPVA. Any 3D confocal reconstruction to show this? If crosslinked droplets were formed, after removing dextran phase, the structure should collapse. Why would the connected pores be seen after swelling in Figure 2D?

Authors' response: Yes. We have 3D confocal imaging data to confirm the stability of PIPS hydrogels. Even if the crosslinked gel phase forms droplets, we observe that the entire hydrogel remains stable and can be handled as a bulk. Similar observations have been made for other microstructured hydrogels, e.g. microfilaments in FLIGHT (Liu et al., Adv Mater, 2022) and micropores in a non-light, PEG-based PIPS system (Zauchner et al., Nature Commun, 2024). We reason that the pore phase might not be completely devoid of any crosslinked nPVA-PEG, but that a few loosely crosslinked polymers might still be present to stabilize the entire structure, while leaving sufficient space for any high-molecular weight tracer perfusion and cell spreading. Due to the difficulty to completely explain the process, we have now removed the images and are showing only a clearly bicontinuous structure (**Revised Fig. 2f**).

Nevertheless, below is a 3D reconstruction of the original droplet-forming composition. It seems evident that the hydrogel phase consists of spherical components, even though the degree of connectivity between the droplets cannot be determined.

Response Figure 2

12- Figure S3a indicates many micropores to me. Why would the authors say that no pores were formed?

Authors' response: We re-evaluated these images in **Figure S3a** and conclude that the signal-to-noise ratio was too low to be identified as micropores. These images are much weaker compared to our other imaging data.

Figure S3a

13- In Figure 2c, both groups contain the same amount of FITC-nPVA before UV. the fluorescence seems to be darker with the 2.5% DS group. why?

Authors' response: When imaging the fluorescence signals of bulk PIPS hydrogels, the signal intensity attenuates more throughout the sample depth, even when using line scanning confocal microscopy. Differences in absolute fluorescence intensity may thus occur when imaging the hydrogel at varying depths above the bottom surface. We thus sought to keep a constant imaging depth in the new confocal imaging experiments (see **Response Figure 1** and **Revised Figure 2f**). Some variation in

fluorescence before UV crosslinking remains, but the different samples still show comparable fluorescence intensities after UV crosslinking (see response to **Reviewer 1 - Question 10**).

Original Figure 2c

14- The authors incubated the porous hydrogel in FITC-dextran for 1-2 days to locate the pores, which seem to be long. Is this duration of incubation necessary? Is the pores are well connected, the penetration of dextran should be fast. Would be helpful to characterize the connectivity of the pores.

Authors' response: For practical reasons, we imaged the samples after overnight incubation and noticed that the signal intensity increases with the overnight incubation, resulting in a higher signal-to-noise ratio. However, we are also able to image the pores already **after 1h of incubation** in tracer solution, as can be seen in **Response Figure 3** below. Additional results on pore connectivity can be found with the next comment.

Response Figure 3

15- The porosity and pores size should also be characterized under different concentrations since the pores are the key point of the work.

Authors' response: We have expanded our analysis of PIPS hydrogels of varying compositions and performed both the RIFFT analysis for pore length scale and new 3D pore analyses (following Vandaele et al., Soft Matter, 2020, 16, 4210-4219) to quantify porosity, pore diameter distribution and pore connectivity in our system. We can quantitatively confirm that porosity and characteristic pore length scales differ significantly between the tested compositions. A Kruskal-Wallis test also showed that the pore diameter distributions differ significantly ($p = 0.0281$) and the follow-up Dunn's multiple comparisons test indicated a statistically significant difference between 3% nPVA with 2.5% vs. 3% DS

($p = 0.0257$). We also used the 3D pore data to quantify pore connectivity and show that most pores are well interconnected, as they have more than one directly adjacent pore (see **Revised Figure 3h-k**)

Revised Figure 3

Reviewer #2 (Remarks to the Author):

The authors report a new hydrogel system based on modified polyvinyl alcohol, dithiol crosslinkers, a water-soluble photoinitiator and dextran sulfate for bioprinting 3D structures with controlled macroporosity. Macroporosity in the printed structures is enabled by photopolymerization-induced phase separation during the printing process and is controlled by means of the light intensity and the content of modified dextran in the system. The material is characterized in terms of its porosity, mechanical properties and turbidity as a function of the concentration of its components and the light intensity used for photocrosslinking. The authors also show that it is possible to use the material for volumetric bioprinting of cell-embedded hydrogel structures.

The paper introduces a PIPS process compatible with live cells and volumetric bioprinting. However, it is not very clear from the paper how this process could significantly improve bioprinting applications. In addition, different materials and different degrees of functionalization of nPVA have been used at different stages of the project, which makes it difficult to compare the characterization results. For example, the mechanical properties of the hydrogels were characterized using hydrogels not containing gelatin, a cell-adhesive peptide, pyrogallol and iodixanol, which were all added to the bioresin used for the bioprinting experiments. In particular, the influence of gelatin (as a sacrificial viscosity enhancer) in the PIPS process and in the macroporosity of the resulting 3D printed matrices should be investigated and shown. Moreover, an experiment that shows how PIPS improves the biological outcome of volumetric bioprinting or that shows a clear application which would not be possible without PIPS process would be highly valuable to improve the manuscript and show its contribution to the field of

biofabrication. Therefore, I cannot accept this paper for publication in NC. Here are some more specific comments:

Author's response: We thank the reviewer for the critical analysis of our work and agree that our PIPS system still faces up-scaling challenges for volumetric bioprinting applications, especially using the gelatin as viscosity enhancer. The reason for our limited capability to explore volumetric bioprinting lies in the large amount of bioresin volume (5-10 mL per experimental group) required for volumetric bioprinting. This requests the large-scale synthesis of all polymer precursors to avoid batch variation.

After careful consideration, we decided that the focus of this manuscript centers on the use of our PIPS compositions for 3D cell culture applications using different cell types, and highlight the feasibility of PIPS-based volumetric printing of acellular constructs as a proof of concept. The results on volumetric bioprinting have been moved to the Supplementary Information instead.

Accordingly, the title of this manuscript has been changed from "~~Photoclick-Phase-separating Hydrogels for 3D Cell culture and Volumetric Bioprinting~~" to "Cell-guiding Microporous Hydrogels by Photopolymerization-induced Phase Separation"

Based on your constructive suggestion, we have performed extensive new 3D cell cultures that demonstrate how PIPS improves biological outcomes. As shown in **Revised Figures 5-7**, our findings suggest that PIPS hydrogels enable faster cell spreading, 3D cell morphogenesis, osteogenic differentiation and extracellular matrix production.

1. In the abstract, the reported hydrogel system is claimed to be "low-cost" (line 36, page 2). Such claims should be avoided when proper evidence is not provided. Proper evidence includes a cost analysis of acquiring the materials and processing them into the precursors of the system and a cost comparison with other potentially similar hydrogel systems available on the market.

Authors' response: Thank you for this comment. We have removed „low-cost,, from the abstract.

2. From the introduction, it is not clear what the potential outlook of this work is. What is being enabled that was not possible before? How is this work specifically contributing to move forward the field of biofabrication?

Authors' response: Thank you for this constructive comment. We have strengthened the manuscript with a future outlook section, Section 3, Page 11, as follows:

"...Integration with perfusion bioreactors may further enhance tissue maturation. Given that the disclosed PIPS hydrogels are not proteolytically degradable, optimization with protease-sensitive motifs may enhance cell-matrix remodeling and 3D cell infiltration..."

3. I suggest the authors include a crosslinking or printing window diagram. This will be helpful to (i) complement the phase diagram and (ii) to visually show the range of concentrations for each component of the hydrogel system where PIPS can happen and a porous crosslinked hydrogel can be obtained. This should be also done for the compositions used for volumetric printing, which were clearly different from the ones used for characterization. Some examples of this can be found in "Naomi Paxton et al 2017 Biofabrication 9 044107".

Authors' response: Thank you for this constructive comment. We have included a window diagram to describe the selection criteria for PIPS: i.e., the initially miscible mix (1 phase) undergoes PIPS upon UV irradiation, forming bicontinuous structures. Since the focus is not on bioprinting

anymore, we decided not to determine the window of resins for printing due to the complexity after adding gelatin to the resin compositions.

Revised Figure 2d-e

4. The purpose of fitting a curve to the pore size vs. light intensity data and providing the equation with the best fit is not clear. Moreover, goodness-of-fit measures such as the standard error of the regression are not provided. The purpose of including this curve fitting in the manuscript should be clearly stated and goodness-of-fit measures should be included along with the equation. Otherwise, such data should not be included in the manuscript.

Authors' response: We have revised the evaluation of light intensity effect on pore size by modeling the pore formation kinetics theoretically, taking into consideration polymer domain growth and coarsening until the onset of gelation. In the **Revised Figure 4**, the data points are now compared to the theoretical fit for the given composition.

Revised Figure 4

We have added the description of the theoretical model, Section 2.2, Page 5, as follows:

“Overall, our data indicate a simple first-order kinetics model for the rate of crosslinking with light intensity $R_c \sim I$. Thus, the characteristic timescale of crosslinking scales with light intensity as $\tau_c \sim R_c^{-1} \sim I^{-1}$. The results confirmed that photocrosslinking is faster at higher light intensities (Fig. 4f). With higher light intensity, the pore morphology has less time to evolve before it is arrested at the onset of the gelation. Assuming diffusive coarsening, the characteristic size of the PIPS domains grows with time as $l \sim t^{1/3}$. Thus, within the characteristic time of crosslinking τ_c the phase separated domains will grow to a size $l \sim I^{-1/3}$, which is the characteristic length scale of the pores. Note that at lower light intensity, when the PIPS domains have more time to grow before gelation, domain growth can transition from diffusive to viscous coarsening. In viscous coarsening, the characteristic size of PIPS domains scales with time as $l \sim t$, which gives $l \sim I^{-1}$. Thus, at lower light intensity we expect a transition for the pore size scaling from $l \sim I^{-1/3}$ to $l \sim I^{-1}$. Kimura et al. investigated the process of PIPS in a mixture of polystyrene and methylacrylate. By changing the light intensity, they obtained a variety of stationary morphologies. Together, these findings suggest that the light intensity can be used to tune the crosslinking dynamics and PIPS, which eventually influences the length scale of the pores.”

5. According to the manuscript, gelatin was added just to enhance the viscosity of the bioresin composition and is not expected to participate in the crosslinking reactions, thereby acting as a sacrificial material. The concentration of gelatin added to the system is high when compared to the main polymer to be crosslinked, which is nPVA (gelatin 5 % vs. nPVA 2 %). How does the presence of gelatin during the photocrosslinking process influence the pore formation and the properties of the final matrix? What proportion of the initially added cells to the bioresin are being washed away with the gelatin from the bioprinted construct when incubating it at 37 °C in cell culture medium? This might be related to the low cell density evidenced in Figs 7 and S13.

Authors’ response: Thank you for your suggestion. When adding gelatin to the resin, changes to the phase separation process were indeed observed and thus the concentration of dextran sulfate (DS) had to be lowered. However, the exact role of gelatin in PIPS remains to be investigated. On one hand, gelatin is a thermo-sensitive component which may enhance the viscosity of the mixture. On the other hand, viscosity has been shown to influence coarsening of droplets during PIPS in a recent study from our group (Zauchner et al. Nature Commun 2024). It is important to mention that making a phase diagram of the gelatin-containing resin (nPVA, DS, gelatin, thiol linker) is beyond our capacity.

In previous work from our group (Qiu et al., Adv Funct Mater 2023), gelatin was added as a viscosity enhancer to non-PIPS PVA hydrogels. Adding sacrificial gelatin leads to a higher initial stiffness. However after washing, around 85% of the gelatin could be removed. For a stiff construct (3% PVA), the change of stiffness is minimal after washing. In the present study, due to the relatively high initial stiffness ($G' = 4.2$ kPa), we expect minor changes to the overall hydrogel stiffness after washing out the gelatin.

From the pore imaging in the presence of cells and the respective pore size quantifications, we also conclude that the pores are not big enough to allow encapsulated cells to be washed out through them. We instead attribute the low cell density to the low initial seeding density (0.5-2 mio cells/mL)

and have increased the cell seeding density to 4 mio cells/mL for all the cellular revision experiments using gel casting.

6. In Table 1, the mechanical properties reported for the hydrogel composition including gelatin (mix 3) were measured before washing out the gelatin, therefore not reflecting the mechanical properties of the final 3D printed construct. The reported values cannot be related to the mechanical properties of the matrix where cells are being cultured for two weeks after the bioprinting process.

Authors' response: As mentioned above, we have now removed the volumetric bioprinting part from the main manuscript. Please also see **reviewer question 2.5** above for our further thoughts on the gelatin-containing resin.

7. Figure S16. The printed structure is not clearly visible. Is it a monolith-like structure? The images from d14 show stronger red staining in the samples without iodixanol. Does this suggest that mineralization decreases when using iodixanol? Can this observation be attributed to some effect of the iodixanol on the bioactivity of the bioresin or to its inherent effect on the optical properties of the material? A hypothesis for this phenomenon should be presented and hopefully investigated. Further phenotypic characterization of the cells cultured within the printed construct (with PIPS vs. without PIPS) might shed light on the possible advantages of the apparent macroporosity conferred to the matrix by PIPS.

Authors' response: As mentioned above, we have now removed the volumetric bioprinting part from the main manuscript due to limited resources to investigate the cellular response in the printed constructs further. Thus, **Figure S16** was also removed. Instead we focus on the use of PIPS for 3D cell culture applications by gel casting.

Reviewer #3 (Remarks to the Author):

Reviewer #4 (Remarks to the Author):

The manuscript entitled, 'Photoclick PIPS Hydrogels for 3D Cell Culture and Volumetric Bioprinting' reported the super-fast volumetric bioprinting of cm-scale macroporous hydrogels with encapsulated 3D hMSC. This article is interesting, but there are major gaps. In its current state, it is not up to the standard of Nature Comm.

Specific comments:

1. Direct fabrication of cell-laden hydrogels using photo-initiated crosslinking of polymers, such as Gelatin Methacryloyl (GelMA) and Polyethylene Glycol Diacrylate (PEGDA), has been reported extensively over 2 decades. The pore size and porosity of these hydrogels are known to support cell

growth. The claim in the introduction, “Existing hydrogel systems ... have very small pore sizes (5 – 50 nm) and fail to provide a permissive environment for encapsulated cells” is wrong. The authors should provide the pros and cons of the reported system in comparison to the existing hydrogel systems for 3D cell culture.

Authors’ response: Thank you for your critical evaluation of this work. We apologize for this inaccurate description of the prior arts and have revised the manuscript Introduction section as follows:

“Ying et al. reported aqueous two-phase systems to engineer microporous polyethylene glycol (PEG)-gelatin methacryloyl hydrogels. Photopolymerization is applied to arrest two distinct phases in an emulsion.”

2. Hydrogel matrices for 3D cell culture must be biodegradable in order to be replaced by the natural ECM produced by the encapsulated cells and to make room for cell proliferation. No biodegradability data is provided in this report.

Authors’ response: We agree with the reviewer that we did not focus on biodegradability in the present study. However, our new 3D cell culture experiments (**Revised Figure 5-7**) demonstrate cells grow, differentiate and produce their own extracellular matrix. Additionally, the PVA matrices are hydrolytically degradable due to the presence of ester linkages. In a follow-up project, we are incorporating matrix metalloproteinase (MMP)-sensitive motifs to the PIPS hydrogel system to improve cell activities. However, it is beyond the scope of this work.

3. Stem cells will only proliferate and differentiate properly in an ECM that mimicks the stem cell niche. The reported system lacks cell binding sites and other biochemical signatures for cell-matrix interaction. As shown in the data, hMSC proliferation is very slow. There is no hMSC phenotype data.

Authors’ response: Thank you for your valuable comment. We included a soluble RGD (CGRGDSP) peptide to the resin mix. This has been shown to be effective to promote cell adhesion for non-PIPS PVA hydrogels in our recent work (Qin et al. Adv Mater 2018; Qiu et al. Adv Funct Mater 2023).

To address the reviewers’ feedback, we have assessed whether the PIPS may influence the localization and conjugation efficiency of soluble RGD. We have therefore optimized the incorporation of the cell-adhesive RGD peptides. We conjugated the RGD peptide directly to nPVA via thiol-ene photoclick conjugation and added this modified nPVA-RGD conjugate to our resins to ensure that the peptides are covalently immobilized in the hydrogel phase. Using this conjugate, we have performed extensive 3D cell culture experiments (see **Revised Figure 5-7**). With the increased cell density and optimized RGD-functionalization, we have noticed a consistent effect on 3D cell morphogenesis within the microporous hydrogel, which can be seen in **Revised Figure 5** and in **Supplementary Figure S9**. Additional results on the phenotypical development of the hMSCs over 28 days of osteogenic culture can be found in the **Revised Figure 6**.

For our answer regarding the low hMSC proliferation, please refer to **reviewer question 1.2**.

4. The authors reported about the alkene functionalization of PVA. The degree of alkene substitution is not clear. Even though the synthesis method has been reported previously, characterization data for the nPVA used in this study should still be provided.

Authors' response: Thank you for pointing this out. The substitution degree of nPVA is 3.5%. We have added the NMR spectra of nPVA and nPVA-RGD conjugates to the **Supplementary Figures 18-19**.

5. Thiol-ene chemistry is sensitive to light and air, which could lead to premature degradation. How has this been taken into consideration?

Authors' response: To the best of our knowledge, photoinitiated radical-mediated thiol-norbornene click chemistry (Hoyle and Bowman, Angew Chem 2010) is the most efficient reaction for *in situ* hydrogel formation. Our group has performed extensive work in applying thiol-ene photoclickable PVA hydrogels for several biomedical applications (Qin et al Adv Funct Mater 2015; Qin et al. Adv Mater 2018; Qiu et al. Adv Funct Mater 2023). Thanks to its high efficiency and tolerance against oxygen inhibition, we are able to cast and 3D-print these hydrogels with excellent stability. For example, we have printed cm-scale hydrogel constructs using 1.5%-3% nPVA with excellent stability and permissiveness for cell growth.

[REDACTED]

Response Figure 4. Examples of thiol-ene hydrogel constructs with superior stability.
(Qiu et al. Adv Funct Mater 2023)

6. The authors mentioned bimodal behavior of the polymer premix. Did they check the turbidity point?

Authors' response: To address the reviewer's question, we focus on the interpretation related to bimodal behavior in phase miscibility. This refers to the presence of two distinct miscibility regions in the phase behavior of the hydrogel premix represented in a binary or ternary phase diagram (including the solvent). Depending on the concentration, the system may either undergo phase separation via mechanisms such as nucleation or spinodal decomposition or remain as a single-phase, miscible solution. In the phase diagram, the turbidity point represents a point on the binodal curve, which separates the single-phase region (homogeneous mixture) from the two-phase region (phase separation).

To investigate this, we determined the turbidity point (cloud point) for a range of polymer concentrations in a simplified system using water as the solvent according to the cloud point method. This method relies on direct optical observation of turbidity to identify the binodal, and the cloud point is the point at which a solution turns and remains turbid during titration. This allowed us to construct an approximate binary phase diagram to characterize the phase behavior under these conditions.

We acknowledge that the hydrogel precursor solution used in our experiments contains PEG-2-SH linkers, PBS instead of water, and LAP photoinitiator, which introduces additional complexity. While the phase diagram was derived from the water-based system, it serves as a qualitative reference for understanding the bimodal behavior and approximate phase boundaries.

Furthermore, we recognize that the turbidity point can also be determined during PIPS, where crosslinking and increasing molecular weight trigger immiscibility of the polymer in the aqueous phase. Although the turbidity points and the phase diagram have not been determined for this process, a

concentration window has been provided above that supports the bimodal behavior of "forming pores via PIPS" or not. The results further indicate that the inclusion of PBS and the photoinitiator in the hydrogel precursor may shift the cloud point slightly due to ionic strength and specific molecular interactions, but the general trends remain consistent.

7. The authors stated that ‘...PIPS resin was compared to a composition with the nonionic dextran instead of DS...’; but there are several reports on the dextran based macroporous gels. Some references are needed to justify this.

Authors’ response: We agree with the reviewer that dextran has been mainly used for the preparation of macroporous hydrogels in literature. However, we reason that the negative charge of nPVA will make an impact on its phase separation from dextran derivatives. We used ionic dextran sulfate (DS) due to its negative charge, which is beneficial for PIPS with nPVA. Our findings show that the DS is indeed more efficient for PIPS than nonionic dextran counterparts with the equal Mw.

Response Figure 5. Structures of nPVA and DS

8. There is no change in water uptake with alkene content, why? In general, degree of gelation is directly related to the crosslinking density and water uptake.

Authors’ response: It seems that the reviewer might be referring to the following statement: "Interestingly, the mass swelling ratio of 2% nPVA hydrogels with varying DS content showed no significant difference (Supplementary Fig. 8)." If we understood correctly, this might be a misunderstanding regarding our use of "DS" as an acronym for "dextran sulfate" and not "degree of substitution". Our alkene content does not change over the characterized compositions and the overall crosslinking density should remain the same. Thus, the aim of this experiment was only to see whether the microarchitectural changes induced by the inclusion of dextran sulfate would have any effect on the mass swelling ratio. Please also see **Supplementary Figure S8:**

Supplementary Figure S8

9. How was the porosity measured? The pore size distribution is not provided either.

Authors' response: Originally, we had only quantified the characteristic pore length scale using the radial intensity of Fast Fourier Transform (RIFFT) method. Please see our new results on porosity and pore size distribution quantification in our answer to **reviewer question 1.15** above.

10. What does 'higher crosslinking density in the nPVA-rich phase' mean? By definition, a hydrogel consists of both solid and liquid phases. The solid phase is crosslinked. Are the authors suggesting that within the "solid" phase of the reported system, there are more than one phases? If so, what are they? And what are the estimated crosslinking densities?

Authors' response: We apologize for this confusion. We agree with the reviewer that a conventional hydrogel indeed consists of both solid and liquid phases. In our PIPS hydrogel, however, the phase distribution seems more complex. We do not mean the solid phase contains more than one phase. Our pore imaging and dye perfusion results show that in the non-hydrogel phase, there are still certain background signals from the labelled nPVA although the pores are perfusable with the tracers. We are interested to study the local crosslinking densities using micro-rheology as reported in literature (Schulz, Anseth et al. PNAS 2015). However, this is beyond the scope of this current study.

We have revised the text, Section 2.3, Page 6, as follows:

"This may lead to a higher crosslinking density in the **hydrogel** phase, resulting in a higher gel stiffness."

Reviewer #5 (Remarks to the Author): I co-reviewed this manuscript with one of the reviewers who provided the listed reports. This is part of the Nature Communications initiative to facilitate training in peer review and to provide appropriate recognition for Early Career Researchers who co-review manuscripts.

REVIEWER COMMENTS

Reviewer #1 (Remarks to the Author):

The authors have included new experiments and data to improve the manuscript. I appreciate the authors' effort. However, some of my previous concerns were not fully addressed.

=> **Authors' Response:** We thank the reviewer for acknowledging the improvements as well as additional suggestions to strengthen this manuscript. We have endeavored to address the comments as much as possible within the scope and time constraints of this manuscript. Please find below a point-by-point response.

1. Novelty and new contribution (following my 1st comment in the previous round).
a. Other phase separation approaches have already proven the benefits of micropores for cell growth. The current manuscript draws the same conclusion using the so-called PIPS. Again, the new contribution is not clear. Is it better than existing ones? Why? I think the other two reviewers also expressed similar concern as the results look so similar to existing literature. The intensively cited reference (Ying et al. Adv Mater 2018) is just an early example. Recently, more and more aqueous two-phase or coacervation microporous hydrogel systems (<https://doi.org/10.1002/adhm.202203243>; <https://doi.org/10.1002/adfm.202400431>; <https://doi.org/10.1002/adma.202300636>; <https://doi.org/10.1002/adma.202410452>) have been reported. The recent work from Tibbitt et al. published in Advanced Materials (<https://doi.org/10.1002/adma.202410452>) actually reported the polymerization-induced phase separation (PIPS) system to introduce pores to hydrogel for 3D cell culture. The fibroblasts (also used in this manuscript) look much more spread than those in this submission. Together, though the authors have included more experiments and data, the core novelty and new contribution of the study are still lacking. The publication of similar PIPS work in other journals further declines the novelty.

=> **Authors' Response:** Thank you for drawing these prior arts to our attention. We have included some of these references to the manuscript (Introduction section).

We agree with the reviewer that these papers represent good alternative examples for microporous hydrogels although they rely on different principles. Importantly, aqueous two-phase (emulsion) systems significantly differ from our photopolymerization-induced phase separation (PIPS) hydrogel system, because it cannot spatiotemporally control pore formation and the resins have phase separated into 2 immiscible phases before photocrosslinking. Despite recent progress, there are a few disadvantages: 1) the emulsion may alter cell distribution and cause uncontrolled cell morphologies in the hydrogels; 2) the emulsion will also result in light scattering and reduction of resolution when used for light-based 3D printing.

In our PIPS hydrogels, the precursor solutions are initially miscible (**one-phase**). When exposed to light, however, they undergo in situ pore formation via PIPS. This process uniquely provides spatiotemporal control of the pore-forming process and resulting microarchitecture of hydrogels, which is not achievable in the aqueous two-phase systems nor in the coacervation system.

As for the paper published in *Advanced Materials* (Dudaryeva et al), thank you for pointing this out. However, as per the scooping policy of *Nature Communications*, it commits to disregard any competing works that are published while a submission to *Nature Communications* is under review or under revision by the authors. More information on this can be found in this editorial - <https://www.nature.com/articles/s41467-020-17817-x>.

To address the reviewer's comment, we have compared the different approaches (principles, materials, advantages, and disadvantages) in **Table 1**.

Table 1. Comparisons of different approaches to fabricate microporous hydrogels.

	Principles	Materials	Advantages	Disadvantages
Aqueous two-phase (Ying et al. Adv Mater 2018)	Emulsification followed by photopolymerization	GelMA/PEG	- Commercial availability of precursors	- No spatiotemporal control of pores - Limited control over cell morphology
Coacervation (Yang et al. Adv Mater 2023)	Supramolecular host-guest complexation	Anthracene/CD-modified PEG and PVA	- Large pores (100 μm) - Cell-compatibility	- No spatiotemporal control of pores
Photopolymerization-induced phase separation (PIPS) (Dudaryeva et al. Adv Mater 2025)	PIPS in viscous resins	nPEG/ dextran /hyaluronan	- Modeling of the kinetics in PIPS - Photo-tuned pore morphology - Enhanced cell migration	- High viscosity of HA and possible cell damage due to high shear stress - No phase diagram nor instructions for PIPS compositions
This work (preprint BioRxiv 2023) Patent disclosure (PCT/EP2023/050516, priority date: 12.01.2022)	PIPS in low-viscous resins	nPVA/ dextran sulfate	- Ease of cell inclusion - Spatiotemporal control - Impact of polymer charge - Enhanced cell viability, differentiation and matrix production - Application of macromolecular crowding - Proof-of-concept 3D printability	- Reliance on custom synthesized precursors (nPVA, nPVA-RGD conjugates)

nPEG, norbornene-functionalized four-arm PEG; nPVA, norbornene-functionalized PVA

To highlight the difference between aqueous two-phase systems and PIPS hydrogels, we have revised the Introduction section as follows:

“Ying et al.⁹ reported aqueous two-phase systems to engineer microporous polyethylene glycol (PEG)-gelatin methacryloyl hydrogels. Although photocrosslinking is applied to arrest two distinct phases in an emulsion, this process does not allow for spatiotemporal control of pore formation.”

b. In the response, the authors explained the differences between aqueous two-phase and PIPS approaches. They stated that the PIPS approach allows for in situ pore formation for 4D tissue and organoid engineering. This statement is not evidenced. If the authors can really identify

some important features for cell culture that the aqueous two-phase system cannot deliver, their work would really make a difference.

=> **Authors' Response:** Thank you for pointing this out. We agree that applying our PIPS hydrogels in 4D tissue and organoid engineering warrants extensive future work. Nevertheless, we are convinced that this study demonstrates a significant advance over the aqueous two-phase system as shown in **Table 1**. The main advantages of our PIPS hydrogel system include low viscosity and ease of cell mixing, tunable pore morphologies, amenable to advanced cell culture techniques (e.g., macromolecular crowding), enhanced stem cell viability and differentiation, and exquisite control of the process (e.g., 3D printability).

2. following my 4th comment in the previous round.

The authors claimed that they did not notice any cell number reduction over the 28 days of culture. Looking at the nuclei (blue DAPI dots) (Figure S9), I think one can see fewer cells on day 28 than on day 3. The authors should double-check such details to avoid misleading.

=> **Authors' Response:** We apologize for the confusion. In our previous response to your 4th comment, we referred to the PIPS group (3% DS) that shows a continuously high cell viability over the first week and no decrease in cell nuclei compared to day 1. We acknowledge that there still seems to be a first increasing, then decreasing trend over the entire time course based on the image quantification of Hoechst-staining data shown in **Response Fig. 1**. As mentioned in our initial response letter and in agreement with previous reports in the literature, cell apoptosis is expected during in vitro osteogenic differentiation as the cells mature, while less cell proliferation is observed. For the control group (0% DS) that shows decreased cell viability, however, the decrease in cell number is consistent with the decreased nuclei count. We attribute this difference in cellular response to the greater accessibility to nutrient transport for cells grown in the PIPS hydrogels compared to cells in the control group.

Response Figure 1. Nuclei count based on Hoechst-stained samples on days 1, 7, 14 and 28 of 3D osteogenic culture (N=3); two-way ANOVA with Šidák's correction for multiple comparisons, *: $p < 0.05$, **: $p < 0.01$. Data presented as mean \pm SD.

3. Photocrosslinking mechanism (following my 9th comment in the previous round)
a. In the original and revised figures (e.g., figure 4c), there is a considerable delay in the increase of modulus in the presence of UV light. In the first tens of seconds, the hydrogel didn't seem to crosslink. what was happening then? Would phase separation happen during this process?

=> **Authors' Response:** Thank you for pointing this out. We like to point out that the delay in the increase of modulus can be attributed to several factors, such as thiol-ene stoichiometric ratio, light intensity, photoinitiator concentration, and the degree of oxygen inhibition. To verify if the phase separation happens or not, we have considered to employ a setup that allows time-lapsed confocal microscopy imaging in combination with in situ shear photo-rheometry. However, we do not have this setup available in our institute. Although it would be extremely interesting, this mechanistic investigation is beyond the scope of this study.

b. Was the delay caused by the PIPS system?

=> **Authors' Response:** Our photo-rheology results (Fig. 2b) show that the addition of dextran sulfate (void-forming agent) did not cause the delay of photocrosslinking.

Figure 2b. Photo-rheology plots of different hydrogel compositions.

c. My previous concern was that the delay might affect the biofabrication process. Volumetric bioprinting claimed super fast fabrication (down to tens of seconds). If the PIPS will only crosslink after tens of seconds of UV irradiation, how to incorporate it with such biofabrication settings?

=> **Authors' Response:** As stated above (comment 3), the photoreactivity of PIPS resins depends on several factors (e.g., thiol-ene ratio, light intensity). The crosslinking efficiency of PIPS resins can be optimized to match the needs for tomographic volumetric printing. One key optimization we have made for volumetric printing is to add gelatin as the viscosity enhancer. It

is important to note that adding gelatin has a major impact on the resin's printability and PIPS process. In our recent work (Qiu et al. Adv Funct Mater 2023), we have demonstrated that adding sacrificial gelatin to thiol-ene PVA resins enhances the photoreactivity and mechanical properties of the hydrogel constructs.

In this study, we focused on applying PIPS hydrogels for 3D cell culture applications - one important area in biofabrication. For volumetric bioprinting, we are currently scaling up the synthesis of PIPS hydrogel precursors (e.g., nPVA, nPVA-RGD conjugates) and cell-expansion process as each print requires at least 1 mL of the bioresins.

Reviewer #2 (Remarks to the Author):

The authors have significantly modified their manuscript taking into account reviewers suggestions and focused more on the material itself and cell culturing and less on 3D bioprinting. I think it was the right decision. The formation of "microporous" hydrogel in-situ in the presence of cells by PIPS is an interesting and important approach, especially for bio applications. Mechanical properties of different materials have been thoroughly investigated and characterized.

The effect of porosity on cells has been also investigated and there is indeed a significant positive effect. Overall, I am willing to recommend the new version of the paper for publication with minor revision. Below are a few minor suggestions for improvement.

One question/suggestion. The field of porous materials has a problem of definitions. Traditionally people separate them into micro- (below 2 nm), meso- (2-50 nm) and macroporous (above 50 nm). This however is not the most convenient nomenclature in my opinion. Nevertheless, I recommend the authors to add one sentence giving their definition of "microporous" materials to avoid possible confusions.

Scale bars (values) are not visible in some images.

=> **Authors' Response:** We thank the reviewer for this positive feedback. To incorporate the suggestions, we have removed scale bar labels within the figures and instead specified all scale bar sizes in the respective figure captions. Additionally, we have revised the introduction to define our distinction between microporosity and nanoporosity as follows:

"Microporous scaffolds¹⁻³ have emerged as promising materials for 3D cell culture and tissue engineering since they facilitate solute transport, cell-cell communication and tissue ingrowth. Methods to create micropores in the range of 1–200 μm include porogen leaching¹⁻³, microgel annealing⁴⁻⁷, microstrand molding⁸, phase separation through emulsification^{9,10} or polymerization^{11,12}, and laser erosion¹³⁻¹⁵. [...] Existing hydrogel systems^{15,18} for 3D bioprinting often have nanopores with very small sizes (5 nm – 100 nm) that fail to provide a permissive environment for encapsulated cells."

Reviewer #4 (Remarks to the Author):

The authors only addressed the reviewer comments partially. Overall, there are not enough new concepts or sufficient novelty to warrant publication in Nature Communication. Even though the authors provided additional data to support porosity and cell-compatibility, they failed to answer the very fundamental question: how is their system/composition significantly better than the existing hydrogel-based scaffolds? Hydrogel, by definition, consists of two phases: solid and water. The use of photopolymerization to produce hydrogels has been reported extensively, including many cell-laden systems with interconnected macropores.

=> **Authors' Response:** Thank you for your constructive comments. We like to emphasize again about the novelty of this work (see **Comment 1, Reviewer 1**). Detailed comparisons with the prior arts are summarized in **Table 1**.

We agree with the reviewer that photopolymerizable hydrogels have been extensively used in the field. The objective of this work is to introduce **photopolymerization-induced phase separation (PIPS)** as a new technique to prepare microporous hydrogels for 3D cell culture and volumetric printing. The beauty of PIPS hydrogels lies in the ability to control pore formation with light and 3D cell-material interactions as well as the capacity to fabricate hierarchically complex hydrogel biomaterials.

As illustrated in **Response Fig. 2**, conventional hydrogels are composed of the solid and water phases where the solid phase has nanoscale pore sizes. However, PIPS hydrogels contain hierarchically structured phases where the microstructured “**hydrogel phase**” is composed of crosslinked polymers and water at the nanoscale, while the “**pore phase**” does not contain the solid gel network.

Response Figure 2. Illustration of the solid/water phases in conventional and PIPS hydrogels.

REVIEWER COMMENTS

Reviewer #1 (Remarks to the Author):

After looking into the revised manuscript and the authors' response (including those to other reviewers), I still hold my point that the work lacks sufficient impact and novelty to be published in Nature Communications.

1-Regarding the novelty and new contribution, I still don't think the current manuscript leads to considerable new knowledge or new approaches.

a. The authors acknowledge the existing competing studies but only include one that was published before their submission date. However, they did cite some newer literature during revision in discussions other than the core novelty session. I think they should make the rule consistent and include the most relevant references, especially when communicating their novelty and contribution.

b. The authors didn't provide evidence but just repeated their statements regarding the 4D tissue and organoid engineering. They don't even intend to change their tone with the claim. This point is associated with the biological outcomes of their material system. Again, they failed to prove the advances of their system over existing phase separation systems regarding cell activities and biofunctionality.

=> **Authors' Response:** We respectfully disagree with the reviewer about the critics on the novelty and new contribution of our work, because 1) we detail the principles of PIPS hydrogels and how one develops such a PIPS hydrogel system (see **Figure 2**); 2) we report a hyaluronan-free (non-viscous) hydrogel system for PIPS, which is significantly different from the PEG-dextran-hyaluronan system which relies on the use of viscous hyaluronan; 3) we also systematically evaluated the physicochemical properties of our PIPS hydrogels as an influence of molecular charge, light intensity, swelling and pore structure for 3D cell culture applications; and 4) we demonstrated the feasibility of PIPS-assisted volumetric printing.

a. We have made the citation of newer literature consistent and also reference the Tibbitt paper while highlighting the key differences (see response to Reviewer 2).

b. Regarding our statement on 4D tissue engineering, we would like to clarify that we meant temporal control over the pore formation process as the 4th dimension in addition to the 3D hydrogel system.

2-The authors seem to realize the conflict between their claim and figure S9, and thus have changed the text. This could be one example, the authors should carefully check the manuscript to make sure they have properly describe their results.

In response Figure 1, more cells were seen in the DS-free control group than the PIPS group at day 1 and day 7. At day 1, the cell number of 3% DS group is just 1/3 of that of 0% DS group. Why? Shouldn't they fix the initial cell amount? does this mean that the PIPS will lose a large number of cells due to the removal of the sacrificial phase? If yes, this is a major drawback.

=> **Authors' Response:** Thank you for your constructive comment. As pointed out by Reviewer 2, the decrease of cell number is quite common during 3D osteogenic differentiation culture. We have carefully checked our experimental methods and results. We are certain that

the initial cell amount is fixed for all groups. We do not think the PIPS group will lose many cells after embedding based on our cell counting results.

3-the delay of photocrosslinking is not clarified properly. They evidenced that the delay was not caused by the PIPS system. But again, how could bioprinting happen within 12s when the first tens of seconds of light irradiation do not induce polymerization? If the addition of gelatin will enhance the photoreactivity, the authors should present the photorheology of the formulations containing gelatin.

=> **Authors' Response:** As we described in the previous response, the photoreactivity of the resin (with gelatin) used in volumetric printing (405 nm) differs significantly from that of the gelatin-free resin as tested in **Figure 4c** (365 nm). At low light intensities, our resins do exhibit the delay of crosslinking which resulted in bigger pore formation.

Please note that the photorheology of gelatin-containing formulations has been included in **Supplementary Figure 16**.

Reviewer #2 (Remarks to the Author):

The authors have satisfactorily addressed all the reviewers' comments and the paper can be published after minor revision. The use of in-situ PIPS to create hierarchically porous hydrogels that enhance cell behavior in 3D cell culture is an important concept, which the authors convincingly demonstrate. The observed decrease in cell number after 28 days of culture is not surprising, but it should be further investigated in future studies using specific cell systems, as this effect is likely to vary depending on the cell type used. The delayed increase in modulus may result from the interplay between phase separation (which is kinetically slower than polymerization) and polymerization. Regarding the paper by Tibbitt and colleagues, it should be properly cited in the manuscript, and any differences or similarities should be clearly and transparently discussed to ensure fairness to the research community.

=> **Authors' Response:** Thank you very much for your constructive comments. We have added the paper from Tibbitt and co-workers and have revised our manuscript as follows:

“...Compared to other hydrogels relying on polymerization-induced phase separation^{12,13}, our PVA-based PIPS hydrogel system provides the following advantages: 1) nPVA has >30 clickable ene groups and offers more flexibility for conjugating bioactive peptide motifs than 4-arm PEG precursors; and 2) it does not rely on the high viscosity of hyaluronan and thus avoids possible cell damage due to high shear stress during mixing...”